# Dietary diversity and evolution of the earliest flying vertebrates revealed by dental microwear texture analysis

Jordan Bestwick [1,2✉], David M. Unwin[3], Richard J. Butler [2] & Mark A. Purnell [1✉]

Pterosaurs, the first vertebrates to evolve active flight, lived between 210 and 66 million years ago. They were important components of Mesozoic ecosystems, and reconstructing pterosaur diets is vital for understanding their origins, their roles within Mesozoic food webs and the impact of other flying vertebrates (i.e. birds) on their evolution. However, pterosaur dietary hypotheses are poorly constrained as most rely on morphological-functional analogies. Here we constrain the diets of 17 pterosaur genera by applying dental microwear texture analysis to the three-dimensional sub-micrometre scale tooth textures that formed during food consumption. We reveal broad patterns of dietary diversity (e.g. *Dimorphodon* as a vertebrate consumer; *Austriadactylus* as a consumer of 'hard' invertebrates) and direct evidence of sympatric niche partitioning (*Rhamphorhynchus* as a piscivore; *Pterodactylus* as a generalist invertebrate consumer). We propose that the ancestral pterosaur diet was dominated by invertebrates and later pterosaurs evolved into piscivores and carnivores, shifts that might reflect ecological displacements due to pterosaur-bird competition.

[1] Centre for Palaeobiology Research, School of Geography, Geology and the Environment, University of Leicester, Leicester LE1 7RH, UK. [2] School of Geography, Earth and Environmental Sciences, University of Birmingham, Edgbaston, Birmingham B15 2TT, UK. [3] Centre for Palaeobiology Research, School of Museum Studies, University of Leicester, Leicester LE1 7RF, UK. ✉email: J.Bestwick@bham.ac.uk; map2@leciester.ac.uk

Pterosaurs, the first group of vertebrates to evolve active flight, lived between 210 and 66 million years ago during the Mesozoic[1]. They had a global distribution[2] and inhabited a range of terrestrial, coastal and marine environments[3]. As important components of Mesozoic biotas[4], robust reconstructions of pterosaur diets are thus crucial not only for understanding the ecological roles they performed within Mesozoic food webs, but for helping address broader ecological and evolutionary debates that enhance our understanding of Mesozoic ecosystems. Key debates include: whether sympatric species of pterosaur competed for, or partitioned, food resources[5]; whether ontogenetic stages of a single species competed for, or partitioned, resources and thus had life-histories more similar to those of extant birds or non-avian reptiles[6–8]; and what impact the emergence and radiation of birds had on pterosaur dietary evolution[9], and vice versa.

Pterosaur diets have been the subject of considerable investigation, but for many taxa there is little consensus regarding their diet[4]. Dietary studies are inherently difficult because pterosaurs have no modern descendants[10], and many dietary hypotheses are based on weakly supported analogies between the morphology and function of anatomical structures in extant animals and pterosaurs[4]. Fossilised stomach contents can be informative but are limited to a small number of specimens from a few species[4,11]. Furthermore, studies that include dietary hypotheses typically focus on just one or two taxa, and the absence of a multi-taxon framework prevents reliable investigations into resource partitioning between sympatric pterosaurs and the role of diet in evolutionary transitions and responses to environmental upheavals[12,13].

A more robust approach to understanding diet involves dental microwear texture analysis (DMTA)—quantitative analysis of the sub-micrometre scale three-dimensional textures that form on tooth surfaces during food consumption[14–17]. The difficulty experienced by consumers in piercing and chewing food items determines the microwear patterns that form on tooth crowns which, consequently, provides direct evidence of diet[18]. Analyses use standardised texture parameters[14,19] to quantify microwear, and thus dietary, differences between species and/or populations and therefore do not assume direct relationships between the morphology and inferred functions of teeth[14,15,17]. Recent work has demonstrated that DMTA of non-occlusal (non-chewing) tooth surfaces of extant reptiles, including archosaurs (the clade to which pterosaurs belong), differs between dietary guilds[18,20]. This relationship between microwear texture and diet in taxa that do not chew food items provides a robust multivariate framework for this study, allowing us to quantitatively test and constrain pterosaur dietary hypotheses, and to explore dietary shifts within this clade across evolutionary time.

Here, we apply DMTA to the non-occlusal tooth surfaces of 17 pterosaur genera. These pterosaurs span the first 120 million years of pterosaur evolution from the Upper Triassic (Norian) to the Upper Cretaceous (Cenomanian). We use DMTA to test three hypotheses: that pterosaur teeth preserve evidence of dietary differences between taxa, between sympatric species and between ontogenetic stages of the same species. Our hypothesis testing uses the multivariate framework provided by DMTA of extant reptiles[18] as well as additional, independent tests using a DMTA framework derived from the non-occlusal tooth surfaces of bats, the only extant group of dentulous flying vertebrates (given the differences between bats, extant reptiles and pterosaurs, consilience in the results of these independent analyses represents a very stringent test of our approach). Our results allow us to propose a number of specific dietary interpretations, including carnivores, piscivores and invertebrates consumers, and to distinguish between dietary specialists and pterosaurs that consumed a range of different prey types. We also provide evidence of sympatric niche partitioning between *Rhamphorhynchus* and *Pterodactylus*, and evidence of ontogenetic niche partitioning in *Rhamphorhynchus*. Furthermore, we reconstruct pterosaur dietary evolution from our DMTA results by projecting pruned, time-calibrated trees from three pterosaur phylogenies into the extant reptile multivariate framework, and by mapping pterosaur microwear characteristics from the reptile multivariate framework onto each tree. We find that the diets of the first pterosaurs largely consisted of invertebrates and that pterosaurs became increasingly carnivorous and piscivorous over evolutionary time. These results provide insight into the multitude of roles that pterosaurs performed within Mesozoic food webs and highlight the applicability of DMTA for investigating extinct ecosystems.

## Results and discussion

**Extant reptile and bat microwear frameworks**. The extant reptile framework comprises six crocodilian and seven monitor lizard species assigned to one of five dietary guilds: carnivores (tetrapod consumers); 'harder' invertebrate consumers (e.g. beetles, crustaceans and shelled gastropods); 'softer' invertebrate consumers (e.g. grasshoppers); omnivores; and piscivores (fish consumers)[18] (Supplementary Fig. 1 and Supplementary Data 1 and 2). See Fig. 1a–f for examples of digital elevation models of extant reptile and pterosaur tooth surfaces from which texture data were acquired. As previously reported[18], four texture parameters differ significantly between reptile guilds (ANOVA; Supplementary Data 3 and 4) and principal components analysis separates them in a texture-dietary space defined by PC axes 1 and 2 (Fig. 2 and Supplementary Fig. 2). PC 1 negatively correlates with proportions of total vertebrates in reptile diets ($r_s = -0.3564$, $P = 0.0004$) but positively correlates with total invertebrates ($r_s = 0.3192$, $P = 0.0016$), while PC 2 positively correlates with dietary proportions of 'softer' invertebrates ($r_s = 0.2907$, $P = 0.0043$; Fig. 2; Supplementary Fig. 3 and Supplementary Data 5). In very broad terms, 'harder' foods correlate with rougher textures, see ref. [18] for details).

The extant bat framework comprises eight species assigned to one of four guilds: carnivores; 'harder' invertebrate consumers; piscivores; and 'softest' invertebrate (e.g. moths) consumers (Supplementary Fig. 1, Supplementary Data 1 and 2 and Supplementary Note 1). PCA of the 15 texture parameters that significantly differ between guilds (Supplementary Data 3 and 4) separates guilds in a texture-dietary space defined by PC axes 1 and 2 (Supplementary Fig. 2). PC 1 positively correlates with proportions of total invertebrates, 'softest' invertebrates, plant matter and dietary generalism, and negatively correlates with total vertebrates and tetrapods. PC 2 positively correlates with dietary proportions of tetrapods (Supplementary Fig. 3, Supplementary Data 5 and Supplementary Note 1).

**Pterosaur dietary reconstructions**. Projecting pterosaur data into the reptile texture-dietary space plots them within the bounds of the reptiles, and where pterosaur taxa are represented by multiple specimens, there is tendency for them to cluster together rather than exhibit a random distribution across the entire texture-dietary space (e.g. *Darwinopterus*, *Dimorphodon*) although some are more broadly distributed (e.g. *Dorygnathus*, *Pterodactylus*; Fig. 2; Supplementary Fig. 2 and Supplementary Note 2). This non-random distribution is noteworthy, given that pterosaur data played no role in structuring the PCA. The independently derived texture-dietary space for bats gives similar results (Supplementary Fig. 3 and Supplementary Note 2). In both analyses, PC 1 explains the majority of microwear variation between reptiles and bats (66.2% and 55% respectively) and is therefore the more

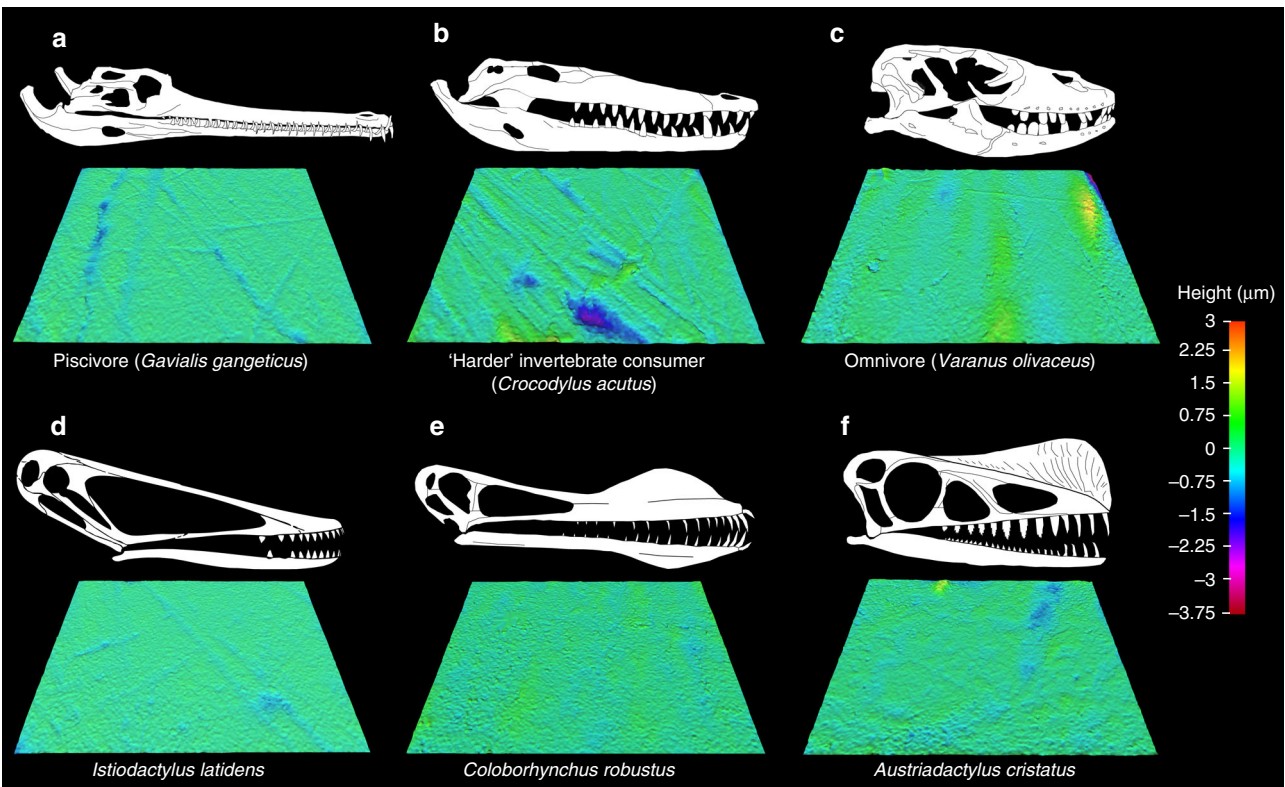

**Fig. 1 Example scale-limited tooth surfaces of reptile dietary guilds and pterosaurs. a–c** Reptile dietary guilds; **a** piscivore (*Gavialis gangeticus*; gharial), **b** 'harder' invertebrate consumer (*Crocodylus acutus*; American crocodile) and **c** omnivore (*Varanus olivaceus*; Grey's monitor lizard). **d–f** Pterosaurs; **d** *Istiodactylus* (PCA plot number 16 in Supplementary Fig. 2), **e** *Coloborhynchus* (PCA number 5) and **f** *Austriadactylus* (PCA number 2). Measured areas 146 × 110 μm in size. Topographic scale in micrometres. Skull diagrams of extant reptiles and pterosaurs not to scale (see 'Methods' for sources).

informative axis for interpreting and constraining pterosaur diets (this does not negate the usefulness of PC 2 as this axis can provide additional constraints, see below). PC 1 approximates a spectrum from invertebrate dominated diets at one end to vertebrate dominated diets at the other. Pterosaur taxa with values similar to vertebrate consuming guilds in the reptile-based analysis also plot with vertebrate consuming guilds in the bat-based analysis. The same is true for those with PC 1 values similar to invertebrate consuming guilds (Fig. 2; Supplementary Figs. 2 and 3 and Supplementary Note 2). That these independent analyses produce similar results provides powerful evidence that the differences between pterosaur non-occlusal tooth textures are attributable to dietary differences. This allows us to test previous dietary hypotheses, provide more refined characterisation of the ecological roles that pterosaurs performed within food webs, and explore their dietary evolution. For simplicity we focus here on the reptile-based analysis (see Supplementary Note 1 for details of the bat-based analysis).

Further support for the robustness of our approach comes from consilience with other, well-supported dietary interpretations. While decades of research have yielded little consensus regarding the diet of most pterosaurs, there are a few taxa for which multiple lines of evidence provide more secure hypotheses. For example, multiple specimens of *Rhamphorhynchus* are preserved in association with fish remains as gut contents, and their sharp conical teeth have been compared to extant piscivores[21–24], a dietary hypothesis with which most analyses agree[4]. In our analysis, all but one of the 14 specimens overlap with reptile piscivores along PCs 1 and 2 (Fig. 2), and the distribution of *Rhamphorhynchus* corresponds more closely to the convex hull for piscivores than any other group (Fig. 2). Similarly, *Istiodactylus* has consistently been reconstructed as a

piscivore or a carnivore[4] based on its razor-edged, lancet-shaped, interlocking teeth, interpreted as an adaptation for defleshing carcasses[25]. Our DMTA results place *Istiodactylus* at the extreme end of the vertebrate scale with respect to reptile-based PC 1, plotting closest to a number of reptile carnivore samples (Fig. 2). This strong concordance between multiple independent lines of evidence gives us confidence in our DMTA results and provides further, quantitative evidence that *Rhamphorhynchus* was a piscivore and that *Istiodactylus* was an obligate vertebrate consumer, most likely a carnivore.

In contrast to the few pterosaur taxa where dietary hypotheses are relatively uncontroversial, DMTA gives us insight into pterosaurs that have poorly constrained diets. *Dimorphodon*, for example, has been variably interpreted as carnivorous, piscivorous and insectivorous based on comparative anatomy of the skull with extant birds and crocodilians (see ref. [4] and references therein). The high degree of overlap of *Dimorphodon* tooth textures with reptile carnivores and piscivores across PCs 1 and 2 (Fig. 2) suggests that the diet of *Dimorphodon* largely consisted of vertebrates, although some consumption of 'softer' invertebrates cannot be entirely ruled out. *Lonchodraco* and *Serradraco* are poorly known pterosaurs whose anatomy and ecology is largely speculative[1,4]. *Serradraco* is located towards the extreme end of the reptile carnivore/ piscivore scale and overlaps more strongly with reptile piscivores along PC 2 (Fig. 2). This suggests that *Serradraco* was predominantly piscivorous. *Lonchodraco* exhibits a very similar PC 2 value to *Serradraco* and a more positive value along PC 1 (Fig. 2) which together suggests the inclusion of some invertebrates within a largely piscivorous diet.

DMTA results challenge some previous dietary hypotheses. *Austriadactylus*, for example, has been interpreted as carnivorous

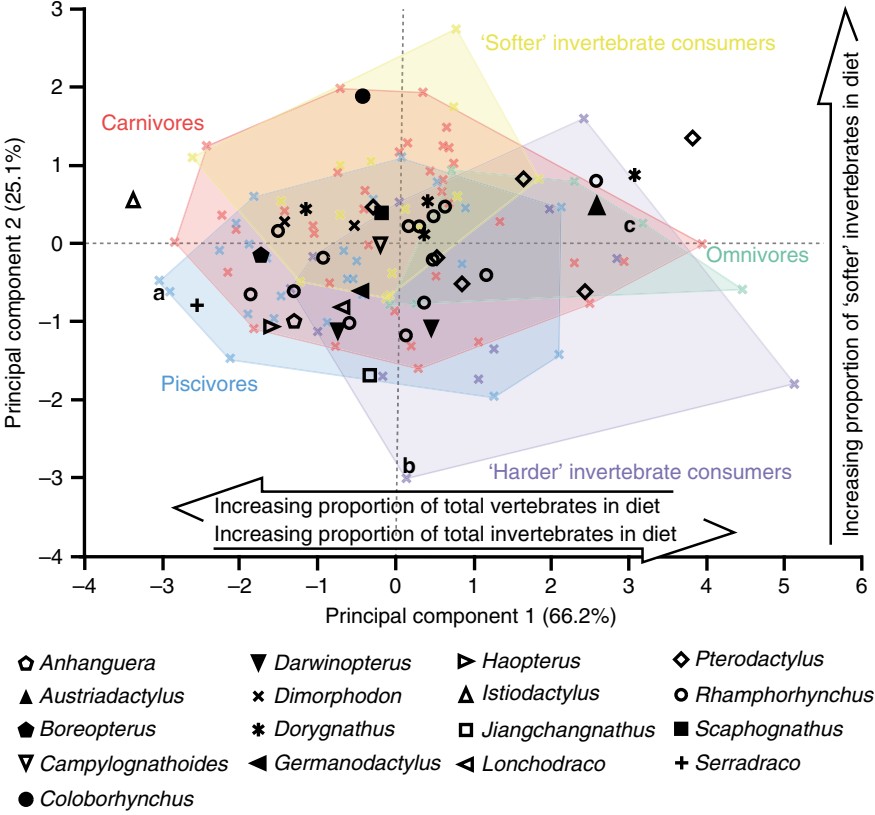

**Fig. 2 Quantitative textural analysis of microwear in extant reptiles and pterosaurs.** Texture-dietary space of International Organisation for Standardisation (ISO) texture parameters for reptiles and pterosaurs. Texture-dietary space based on extant reptile data ($n = 95$) with pterosaurs ($n = 40$) projected onto the first two axes as unknown datum points. Extant specimens with associated letters represent surfaces **a–c** in Fig. 1. Arrows show significant correlations of dietary characteristics along PC axes 1 and 2. Pterosaur specimen information corresponding to PCA plot number can be found in Supplementary Fig. 2 and in Supplementary Data 1. Texture-dietary space adapted from Fig. 2 of ref. [18] (https://www.nature.com/articles/s41598-019-48154-9/figures/2) by Jordan Bestwick and Mark Purnell under a Creative Commons Attribution 4.0 International License https://creativecommons.org/licenses/by/4.0/ to include the pterosaur data.

based on functional analyses of the closing mechanics of its jaws[26]. However, its position in the texture-dietary space, with high values for PC 1, suggests a diet dominated by invertebrates (Fig. 2). *Coloborhynchus* has been interpreted as piscivorous based on its tooth morphology[24] and on theoretical modelling of possible fishing behaviours[27]; in the texture-dietary space its PC 1 value is comparable to carnivores and piscivores but its positive PC 2 value suggests it also consumed a high proportion of 'softer' invertebrates (Fig. 2). It is unlikely that *Coloborhynchus* was a specialist piscivore; our analysis points towards a broader dietary range than has previously been proposed.

**Sympatric partitioning and ontogenetic dietary shifts**. Our results also have a bearing on contentious hypotheses of ecological interaction and competition between taxa: did sympatric pterosaurs compete for or partition food resources[5,24,28]? DMTA demonstrates niche partitioning between pterosaurs from the Solnhofen Limestones. *Rhamphorhynchus* and *Pterodactylus*, taxa with the largest sample sizes, exhibit separation along PC 1 (reptile PCA: $t = -2.431$, d.f. = 18, $P = 0.0257$; Fig. 2; see Supplementary Note 2 for similar results in the bat-based analysis), but not along PC 2 ($t = -1.303$, d.f. = 18, $P = 0.2087$). More negative PC 1 values for *Rhamphorhynchus* indicate piscivory, while *Pterodactylus* individuals exhibit a broader range of more positive PC 1 values, indicating invertebrate-dominated, and possibly more generalist diets (Fig. 2). This agrees with theoretical models of pterosaur feeding behaviours that suggest *Pterodactylus*

had relatively strong bite forces which would have facilitated dietary generalism[29]. It is difficult to determine unequivocal dietary preferences for Solnhofen species represented by single specimens that fall in the centre of the texture-dietary space, but *Scaphognathus* and *Germanodactylus* may have consumed both vertebrates and invertebrates (Fig. 2), with evidence from the analysis of bat diets suggestive of more invertebrates (Supplementary Fig. 3). While this might indicate finer niche partitioning between Solnhofen pterosaurs, it is also likely that these species were not typical members of the community as they are relatively rare (<5 specimens in both cases).

Ontogenetic dietary shifts have been suggested for several pterosaurs[6–8,30,31] but the hypothesis lacks evidential support. In our study, the dietary separation of *Rhamphorhynchus* specimens along PC 1 correlates with specimen size (reptile PCA; lower jaw length; $r_s = -0.6044$, $P = 0.0221$; Fig. 3 and Supplementary Note 2). PC 2 values do not correlate with size ($r_s = -0.3451$, $P = 0.2269$). Specimens that most likely represent highly immature individuals based on their size and degree of skeletal ossification[6] exhibit positive PC 1 values, indicating invertebrate-dominated diets. Specimens that represent late-stage juveniles and adults exhibit decreasing PC 1 values and thus increasingly piscivorous diets (Fig. 2). Ontogenetic dietary shifts are common in extant reptiles where offspring usually feed themselves (e.g. ref. [32]), but are not exhibited by birds[8]. DMTA thus provides direct evidence of ontogenetic dietary shifts in a pterosaur, and supports the hypothesis that pterosaur life-histories were more like those of reptiles than birds or bats[6–8,30].

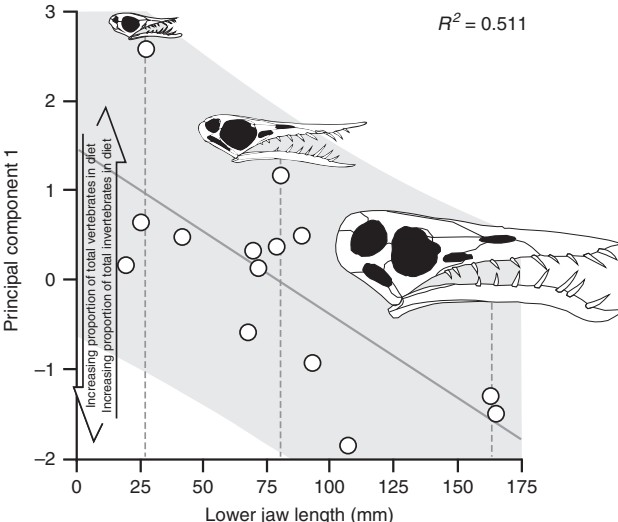

**Fig. 3 *Rhamphorhynchus* ontogenetic shifts in the reptile texture-dietary space.** Regression of lower jaw length ($n = 14$), as a proxy for specimen size, plotted against PC 1 values from the extant reptile texture-dietary space. The solid grey line denotes the line of best fit and grey shading denotes the 95% confidence intervals. Skull diagrams represent scaled examples of an adult, juvenile and hatchling *Rhamphorhynchus*, with the grey dashed lines denoting their relative size. These diagrams illustrate the size disparity between sampled life-history stages and are not specimen reconstructions of the datum points they are associated with. See Supplementary Fig. 4 for *Rhamphorhynchus* ontogenetic shifts in the bat texture-dietary space and the 'Methods' for the skull diagram sources.

**Pterosaur dietary evolution**. The implications of our results for pterosaur dietary evolution were explored in two ways. First, by independently projecting pruned, time-calibrated trees from three pterosaur phylogenies, Lü et al.[33], Andres and Myers[34], and Wang et al.[35] into the first two axes of the reptile texture-dietary space, creating 'phylo-texture-dietary spaces' analogous to phylomorphospaces. Second, by mapping pterosaur PC scores from the first two reptile texture-dietary axes onto each tree. The results using the Lü et al.[33] phylogeny (Fig. 4, Supplementary Figs. 5, 6 and 9 and Supplementary Data 6) are presented here as this phylogeny exhibits higher stratigraphic congruence than alternatives[36] (see Supplementary Figs. 7 and 10, Supplementary Data 6 and Supplementary Note 3 for results using the Andres and Myers[34] phylogeny, and Supplementary Figs. 8 and 11, Supplementary Data 6 and Supplementary Note 3 for results using the Wang et al.[35] phylogeny). PC 1 values of the pterosaur specimens included for dietary evolution reconstructions negatively correlate with stratigraphic age ($r_s = -0.4859$, $P = 0.048$) and PC 2 values do not correlate with age ($r_s = 0.0552$, $P = 0.8333$). There is a significant phylogenetic signal in pterosaur microwear along PC 1 ($K = 0.9368$, $P = 0.031$, $\lambda = 0.9999$, $P = 0.000143$), but no signal along PC 2 ($K = 0.262$, $P = 0.775$, $\lambda = 0.201$, $P = 0.4014$). The ancestral PC 1 value estimate of Pterosauria (node 1, Supplementary Fig. 9) is 2.2 (Fig. 4b), and the ancestral PC 2 estimate is 0.428 (Supplementary Fig. 5 and Supplementary Data 6). This suggests that the ancestral pterosaur diet was invertebrate-dominated. Over time, pterosaurs move through phylo-texture-dietary space to occupy increasingly negative PC 1 values, reflecting a shift towards consumption of more vertebrates. This is most pronounced in members of Istiodactylidae and Lonchodectidae (nodes 13 and 16 respectively, Supplementary Fig. 9), which have estimated ancestral PC 1 values of $-2.137$ and $-2.007$, respectively (Supplementary Data 6) and independently evolved into obligate vertebrate

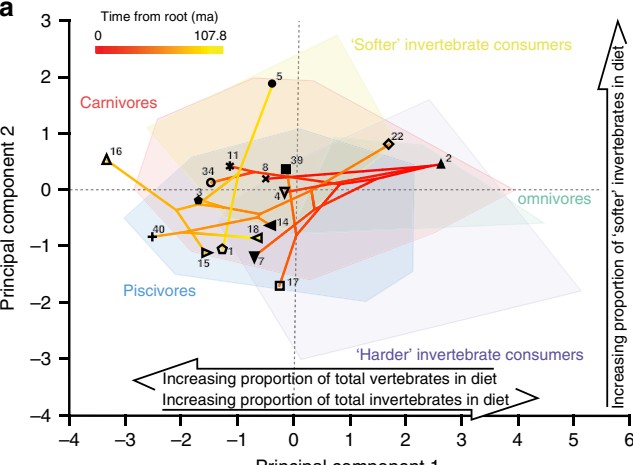

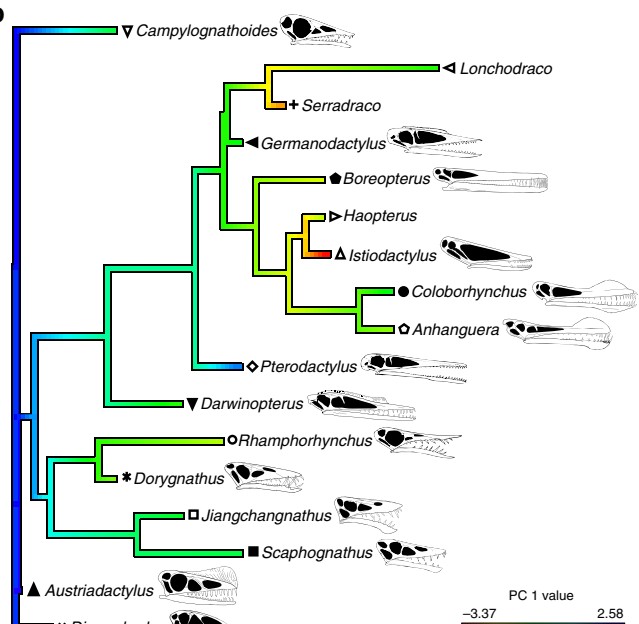

**Fig. 4 Pterosaur dietary evolution. a** Phylo-texture-dietary space of pterosaur microwear from projecting a time-calibrated, pruned tree from Lü et al.[33] onto the first two PC axes of the extant reptile texture-dietary space. **b** Ancestral character-state reconstruction of pterosaur dietary evolution from mapping pterosaur PC 1 values onto a time-calibrated, pruned tree from Lü et al.[33]. To account for ontogenetic changes in diet, only the largest specimen of respective pterosaur taxa, identified by lower jaw length, were included. Pterosaur symbols same as Fig. 2. Skull diagrams of well-preserved pterosaurs not to scale (see 'Methods' for sources). Phylo-texture-dietary space adapted from Fig. 2 of ref. [18] (https://www.nature.com/articles/s41598-019-48154-9/figures/2) by Jordan Bestwick under a Creative Commons Attribution 4.0 International License https://creativecommons.org/licenses/by/4.0/ to remove the extant reptile datum points and to include the pterosaur data and phylogeny.

consumers (Fig. 4b). Interestingly, the invertebrate-dominated diet of *Pterodactylus* and the possible inclusion of invertebrates in the diet of *Lonchodraco* are both secondarily derived (Fig. 4b).

These results provide quantitative evidence that pterosaurs initially evolved as invertebrate consumers before expanding into piscivorous and carnivorous niches[12,13]. The causes of this shift towards vertebrate-dominated diets require further investigation, but might reflect ecological interactions with other taxa that radiated through the Mesozoic[12,13]. Specifically, competition with

birds, which first appeared in the Upper Jurassic and diversified in the Lower Cretaceous, has been invoked to explain the decline of small-bodied pterosaurs, but this hypothesis is controversial[5,9,30,37]. DMTA provides an opportunity for testing hypotheses of competitive interaction upon which resolution of this ongoing debate will depend.

In summary, our analyses provide quantitative evidence of pterosaur diets, revealing that dietary preferences ranged across consumption of invertebrates, carnivory and piscivory. This has allowed us to explicitly constrain diets for some pterosaurs, enabling more precise characterisations of pterosaurs' roles within Mesozoic food webs and providing insight into pterosaur niche partitioning and life-histories. Our study sets a benchmark for robust interpretation of extinct reptile diets through DMTA of non-occlusal tooth surfaces and highlights the potential of the approach to enhance our understanding of ancient ecosystems.

## Methods

**Specimen material**. Tooth microwear textures were sampled from 17 pterosaur genera. Eight bat species were sampled and the microwear texture data of 13 reptile species, comprising six crocodilians and seven monitor lizards, from ref. [18] were included to serve as extant multivariate frameworks. Crocodilians and monitor lizards were chosen to provide the primary framework because their dentitions have been compared with those of pterosaurs[12], and they often forage on the margins of terrestrial and aquatic environments[38,39], comparable to hypothesised pterosaur foraging behaviours[40]. Bats were chosen as an independent comparator framework because they are the only extant group of dentulous flying vertebrates, and their foraging behaviours have sometimes been extrapolated to pterosaurs (see ref. [1] and references therein). Extant and fossil specimens were sampled from the Bayerische Staatssammlung für Paläontologie und Geologie, Munich, Germany (BSPG); Field Museum of Natural History, Chicago, USA (FMNH); British Geological Survey, British Geology Collections, Keyworth, UK (GSM); Institute of Vertebrate Paleontology and Paleoanthropology, Beijing, China (IVPP); Grant Museum of Zoology, University College London, UK (LDUCZ); Museum für Naturkunde der Humboldt-Universität Berlin, Berlin, Germany (MB); Natural History Museum, London, UK (NHMUK); University of Oxford Museum of Natural History, Oxford, UK (OUMNH); Paleontological Museum of Liaoning, Shenyang, China (PMOL); Staatliches Museum für Naturkunde, Karlsruhe, Germany (SMNK); Staatliches Museum für Naturkunde, Stuttgart, Germany (SMNS); Teylers Museum, Haarlem, Netherlands (TM); Royal Tyrrell Museum of Palaeontology, Drumheller, Canada (TMP); Florida Museum of Natural History, Gainesville, USA (UF); and the National Museum of Natural History, Smithsonian Institute, Washington DC, USA (USNM). See Supplementary Data 1 for the complete specimen list.

**Dietary guild assignments**. Reptiles and bats were selected to include taxa with well-constrained dietary differences determined from stomach and/or faecal content studies[38,41–60] (Supplementary Data 2). Studies were chosen that met as many of the following criteria as possible: representative sample sizes, dietary compositions represented as volumetric data (or frequency data at an absolute minimum); and spatial proximity of the dietary study to the location(s) from which the museum specimens we analysed were collected (specimen provenance data, where known, is included in Supplementary Data 1). Taxa that had not been subjected to ecological studies that provided volumetric or frequency breakdowns of diet were not sampled for study. Many reptiles exhibit ontogenetic shifts in diet[32,41,45,48], thus similarly sized specimens from extant species were sampled where possible to minimise effects of this confounding variable. Lower jaw lengths, measured in lateral view, were used as a proxy for overall body size. For extant reptiles, lower jaw lengths were measured from the anterior tip of the dentary to the posterior margin of the surangular. Bat jaw lengths were measured from the anterior to posterior tip of the dentary. Specimen availability enabled *Crocodylus porosus* to be sampled as two separate life-history groups for analysis. Specimens with lower jaw lengths of less than 50 cm were classified as juveniles; specimens with jaw lengths exceeding 50 cm were classified as adults[18,61].

Extant reptiles and bats were assigned to dietary guilds based on the relative 'intractability' (roughly equivalent to hardness) of prey as food[18]. Guilds include carnivores (tetrapod consumers); piscivores (fish consumers); 'harder' invertebrate consumers (invertebrates with hard exoskeletons, e.g. beetles, crustaceans and shelled gastropods); 'softer' invertebrate consumers (invertebrates with less hard exoskeletons, e.g. crickets, grasshoppers, dragonflies, damselflies and ants); 'softest' invertebrate consumers (invertebrates with soft exoskeletons, e.g. invertebrate larvae, butterflies, moths, arachnids and millipedes); omnivores (combination of plant and animal matter). These guilds do not represent the entire dietary diversity of the extant clades, but were selected because of their ecological relevance to pterosaur diet. For example, many New World bat species are nectarivores, sanguivores (blood feeders) or obligate frugivores[62]. These diets have never been

proposed for dentulous pterosaurs[4]. In reptiles, *Crocodylus porosus* adults (saltwater crocodile, $n = 6$), *Varanus komodoensis* (Komodo dragon, $n = 4$), *Varanus nebulosus* (clouded monitor, $n = 11$), *Varanus rudicollis* (roughneck monitor, $n = 8$), and *Varanus salvator* (Asian water monitor, $n = 8$) were assigned to the carnivore guild (total $n = 37$); *Crocodylus acutus* (American crocodile, $n = 7$) and *Crocodylus porosus* juveniles ($n = 5$) were assigned to the 'harder' invertebrate consumer guild (total $n = 12$); *Varanus niloticus* (Nile monitor, $n = 8$) and *Varanus prasinus* (emerald tree monitor, $n = 7$) were assigned to the 'softer' invertebrate consumer guild (total $n = 15$); *Varanus olivaceus* (Gray's monitor, $n = 6$) was assigned to the omnivore guild (total $n = 6$); and *Alligator mississippiensis* (American alligator, $n = 8$), *Caiman crocodilus* (spectacled caiman, $n = 6$), *Crocodylus niloticus* (Nile crocodile, $n = 4$), and *Gavialis gangeticus* (gharial, $n = 7$) were assigned to the piscivore guild (total $n = 25$). In bats, *Trachops cirrhosis* (fringe-lipped bat, $n = 8$) and *Vampyrum spectrum* (spectral bat, $n = 9$) were assigned to the carnivore guild (total $n = 17$); *Nyctalus noctula* (noctule bat, $n = 7$) and *Rhinolophus ferrumequinum* (greater horseshoe bat, $n = 8$) were assigned to the 'harder' invertebrate consumer guild (total $n = 15$); *Myotis capaccini* (long-fingered bat, $n = 7$), *Myotis mystacinus* (whiskered bat, $n = 6$), and *Plecotus auritus* (brown long-eared bat, $n = 6$) were assigned to the 'softest' invertebrate consumer guild (total $n = 19$); and *Noctilio leporinus* (greater bulldog bat, $n = 8$) was assigned to the piscivore guild (total $n = 8$). See Fig. S1 for an overview of how reptiles and bats were assigned to guilds. The *Cr. acutus* skull diagram in Fig. 1 was drawn from UF 54201, the *G. gangeticus* diagram from Wikimedia Commons under the Creative Commons Attribution-Share Alike 2.0 Generic license (https://creativecommons.org/licenses/by-sa/2.0/deed.en), and the *V. olivaceus* diagram from UF 57207.

**Sampling strategy**. Pterosaur specimens were cleaned before sampling using an ethaline solvent gel, produced, and applied according to Williams and Doyle[63]. Extant reptile and bat teeth from dry skeletal specimens were cleaned using 70% ethanol-soaked cotton swabs to remove dirt and consolidant. Microwear data were acquired from non-occlusal (non-chewing) labial surfaces, as close to the tooth apex as possible. In extant reptiles, the mesial-most dentary tooth was sampled; in pterosaurs, the mesial-most tooth of the premaxilla or dentary tooth, based on preservation quality, was sampled. No preference was given to the left or right tooth in extant reptiles and teeth were pooled in analyses. Wear facets that likely formed from tooth–tooth occlusion from the opening and closing of jaws, characterised by their vertical orientation, elliptical shape and parallel features[12,64], were not sampled. Bat microwear data were acquired from the non-occlusal labial surface of the canine in the dentary, as close to the apex as possible. Canines were sampled over premolars and molars because they represent closer functional analogues to reptile teeth since they are not used for chewing. No preference was given to the left or right canine. To test for ontogenetic dietary shifts in *Rhamphorhynchus*, lower jaws were measured in lateral view from the anterior tip of the dentary to the posterior margin of the surangular. Lower jaw length is a reliable proxy for overall specimen size as *Rhamphorhynchus* skulls exhibit strongly linear size relationships with respect to other anatomical structures, such as the humerus and wing finger[6], and we follow this analysis in our use of 'adult', 'juvenile' and 'hatchling'. The example adult and hatchling *Rhamphorhynchus* skull diagrams in Fig. 3 were redrawn from Bennett[6], and the example juvenile diagram was traced from ref. [4] under a Creative Commons Attribution 4.0 International License (https://creativecommons.org/licenses/by/4.0/). High fidelity moulds were taken of teeth using President Jet Regular Body polyvinylsiloxane (Coltène/Whaledent Ltd., Burgess Hill, West Sussex UK). Initial moulds taken from each specimen were discarded to remove any remaining dirt and all analyses were performed on second moulds. Casts were made from these moulds using EpoTek 320 LV Black epoxy resin mixed to the manufacturer's instructions. Resin was cured for 24 h under 200 kPa (2 Bar/30 psi) of pressure (Protima Pressure Tank 10L) to improve casting quality. Small casts were mounted onto 12.7 mm SEM stubs using President Jet polyvinylsiloxane with the labial, non-occluding surfaces orientated dorsally to optimise data acquisition. All casts were sputter coated with gold for three minutes (SC650, Bio-Rad, Hercules, CA, USA) to optimise capture of surface texture data. Replicas produced using these methods are statistically indistinguishable from original tooth surfaces[65].

**Surface texture data acquisition**. Surface texture data acquisition follows standard laboratory protocols[16–18,65,66]. Data were captured using an Alicona Infinite Focus microscope G4b (IFM; Alicona GmbH, Graz, Austria; software version 5.1), using a ×100 objective lens, producing a field of view of 146 × 100 μm. Lateral and vertical resolution were set at 440 and 20 nm, respectively. Casts were orientated so labial surfaces were perpendicular to the axis of the objective lens.

All 3D data files were processed using Alicona IFM software (version 5.1) to remove dirt particles from tooth surfaces and anomalous data points (spikes) by manual deletion. Data were levelled (subtraction of least-squares plane) to remove variation caused by differences in tooth surface orientation at the time of data capture. Files were exported as .sur files and imported into Surfstand (software version 5.0.0 Centre for Precision Technologies, University of Huddersfield, West Yorkshire, UK). Scale-limited surfaces were generated through application of a fifth-order robust polynomial to remove gross tooth form and a robust Gaussian filter (wavelength $\lambda_c = 0.025$ mm)[17,67]. ISO 25178-2 areal texture parameters[19]

were then generated from each scale-limited surface. Descriptions of ISO parameters can be found in Supplementary Data 4 (from ref. [18]).

**DMTA statistical analyses.** Log-transformed texture data were used for analyses as some of the texture parameters were non-normally distributed (Shapiro-Wilk, $P >$ 0.05). The parameter Ssk was excluded from analysis as it contains negative values and thus cannot be log-transformed. To test the hypotheses that microwear differs between reptiles and bats from different dietary guilds, analysis of variance (ANOVA) with pairwise testing (Tukey HSD) was used for each texture parameter. Where homogeneity of variance tests revealed evidence of unequal variances, Welch analysis of variance was used. Principal component analysis (PCA) was used to explore texture parameters exhibiting significant differences between reptile and bat dietary guilds. ANOVA with pairwise testing was used on the PC axis 1 and 2 values of dietary guilds to determine whether guilds occupy different areas of multivariate space along these axes (see ref. [18] for results for reptiles). To test the hypotheses that reptile and bat microwear differences are determined by dietary differences, we used Spearman's rank to test for correlations between PC axes 1 and 2 and dietary characteristics. Additional analyses were employed to independently test for subtle microtextural differences between dietary guilds. Texture parameters were ranked by guild based on the average value for the guild; matched pairs $t$-tests were used to compare the profiles of average parameter values between guilds (see ref. [18] for results for reptiles, and Supplementary Tables 1 and 2 for results for bats).

A Benjamini–Hochberg (B–H) procedure was used to account for the possibility of inflated Type I error rates associated with multiple comparisons[68]. The false discovery rate was set at 0.05. The B–H procedure was not needed for the Tukey HSD tests as it already accounts for inflated Type I error rates[69].

Pterosaur microwear data were independently projected onto the axes of the reptile and bat analyses. Correlation of *Rhamphorhynchus* lower jaw lengths with PC 1 values were tested using Spearman's rank tests. All DMTA analyses were performed with JMP Pro 12 (SAS Institute, Cary, NC, USA) except for the B–H procedure[70], which used Microsoft Excel (www.biostathandbook.com/multiplecomparisons.html).

**Pterosaur dietary evolution.** Pterosauria is defined as the most recent common ancestor of *Preondactylus buffarinii* and *Quetzalcoatlus northropi* and all its descendants[4]. Several competing phylogenies exist for Pterosauria which show similar, but disputed, taxonomic contents of clades and grades of morphologically similar taxa[4]. We therefore used three independent phylogenies, Lü et al.[33], Andres and Myers[34] and Wang et al.[35], to visualise reconstructed pterosaur diets from DMTA in evolutionary contexts. These phylogenies have extensive taxonomic sampling across Pterosauria and contain the majority of taxa in this study. See Supplementary Note 3 for Andres and Myers[34] and Wang et al.[35] results. A time-calibrated tree was constructed from each phylogeny using the DatePhylo function of the R package strap, version 1.4 (ref. [71]) (R version 3.2.3 (ref. [72])). First and last appearance dates were obtained from Dean et al.[3] and the Paleobiology Database (PDBB: www.paleobiodb.org). Branches were scaled using the 'equal' method which eliminates zero-length branches by sharing duration from the more basal non-zero-length branches[71]. Time-scaled trees were pruned in R to include only pterosaurs subjected to DMTA. Pruning has no effect on the topological relationships of remaining pterosaurs. Missing pterosaurs (i.e. species included in the DMTA dataset but not in the phylogenetic hypothesis) were included by hand prior to time-scaling based on information from the literature on their likely phylogenetic position (e.g. assignments to particular clades in published systematic assessments). To minimise confounding effects of ontogeny, only the largest specimen of each pterosaur taxa was included in the analysis, identified by lower jaw length in lateral view from the anterior tip of the dentary to the posterior margin of the surangular. Each tree was independently projected onto the first two PC axes of the reptile texture-dietary space to generate pterosaur phylo-texture-dietary spaces using the phylomorphospace function of the R package phytools, version 0.6-60 (ref. [73]). Phylogenetic signals in pterosaur microwear along PCs 1 and 2 were tested using the phylosig function in phytools[73]. Pterosaur PC 1 and PC 2 values were compared to time from the root of the time-calibrated trees in millions of years to test for shifts through phylo-texture-dietary space over evolutionary time. Ancestral character-state reconstructions of pterosaur diets were performed by independently mapping pterosaur PC 1 and PC 2 values as continuous characters onto each pruned tree using the contMap function in phytools[73]. Ancestral PC 1 and PC 2 values (and the variance and 95% confidence intervals) were calculated for each node within the three phylogenies. Pterosaur skull diagrams in Figs. 1 and 3–4 and Supplementary Figs. 4 and 6–8 were traced from ref. [4] under a Creative Commons Attribution 4.0 International License (https://creativecommons.org/licenses/by/4.0/) except *Anhanguera*, *Austriadactylus*, *Boreopterus*, *Coloborhynchus* and *Jiangchangnathus*, which were redrawn from refs. [74–78], respectively.

**Reporting summary.** Further information on research design is available in the Nature Research Reporting Summary linked to this article.

## Data availability
All the data analysed in this study are publically available in Zenodo (https://doi.org/10.5281/zenodo.4018876)[79] and within the Supplementary Information files.

## Code availability
The code used for the dietary evolution reconstructions can be found in the R file Supplementary Code 1, and is available in Zenodo (https://doi.org/10.5281/zenodo.4018876)[79].

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

## Acknowledgements

Thanks to P. Campbell, M. Carnall, T. Davidson, E. Frey, S. Harris, J. Hay, S.-X. Jiang, D. Lunde, R. Miguez, O. Rauhut, A. Resetar, R. Schoch, T. Schossleitner, A. Schulp, C. Sheehy, L. Steel, B. Strilisky, A. Wynn and C.-F. Zhou for specimen access. Thanks to S. Gabbott and D. Henderson for comments. This work was funded by an NERC studentship awarded through the Central England NERC Training Alliance (CENTA; grant reference NE/L002493/1) and the University of Leicester, and by a Palaeontological Association Sylvester-Bradley Award (PA-SB201701) to J.B. J.B. was also supported by a Leverhulme Trust Research Project Grant (RPG-2019-364) during the completion of the project.

## Author contributions

M.A.P. and D.M.U. conceived the study. All authors contributed to the analytical design. J.B. collected the data. J.B. analysed the data with input from M.A.P. and R.J.B. J.B. wrote the paper with contributions from all authors.

## Competing interests

The authors declare no competing interests.
