## [Peer Review File · Nature Communications]

Reviewers' Comments:

Reviewer #1:

Remarks to the Author:

This manuscript presents the first application of DMTA to pterosaurs to reconstruct their debated diets. Although pterosaurs just represent a new application of a well-established method (DMTA), it still greatly expands the field of applications. To do so, the manuscript builds on databases of extant reptiles (already published) and bats (new). This is the novelty of this manuscript.

This study will therefore be of interest to the DMTA and extinct reptile communities. My review focuses on the content and soundness of the manuscript. I will leave it up to the editor to decide whether or not it has a sufficient potential impact for Nature Communications.

The manuscript is well-written and the argument easy to follow.

Beside some minor comments (see below for details), I have two important issues:

- The authors discuss almost exclusively the results of PC1. They should take PC2 into account too.

This would give a more accurate (and nuanced) presentation and interpretation of the results. As is, some interpretations seem to over-interpret the data. This is true for the main text and for SI text as well. See also comments #15-17 & 23 below.

- A lot of the raw data are missing. As is, the study is not reproducible. I believe this is fundamental for any publication, and especially for Nature Research / Springer Nature. See also comments #6, 8 & 11 below.

Introduction

1) Line 50: the impact of the evolution of pterosaurs on emergence and radiation of birds would be of interest too.

2) Lines 66-68: please also cite Winkler et al. 2019 here.

3) Lines 68-71: the data show that it works, but I am still wondering how DMTA can work if there is no prolonged contact between food and teeth. This is not addressed in Bestwick et al. 2019. Could the authors elaborate here (or in the discussion)? I think this is a very important aspect of this application to reptiles.

Methods

4) Lines 209-255: it is not clear here what comes from Bestwick et al. 2019 and what is new. It should be emphasized. See also comment #20.

5) Lines 256-257: sample size is missing. It should not be necessary to look in SI and count manually to get sample sizes of the important material. As a comparison, there are more details about the reptile dataset, even though it is already published.

6) Lines 265-266: how good is the correlation of jaw length with size, and age? This is a critical aspect before using it as a proxy for size/age. Additionally, how do the authors define "most immature individuals" and "late-stage juveniles and adults" (lines 169-170)? Finally, where are the data?

7) Line 279: although not quite critical for focus variation, please report the NA value of the 100x objective.

8) Lines 283-291: why not use (or at least adapt) Arman et al. (2016)'s template? For example, the thresholding/removing outliers steps seem necessary for the present analysis: judging from Figs 1 and S3 (and the color scales), I would expect high spikes/outliers to be present. Analyzing without excluding these aberrant points is dangerous, especially for absolute height parameters (Sz, Sv, Sp...). To increase reproducibility, the authors should consider uploading their 3D data files (the best being MNT files including the surface, the analysis workflow and the results) on Dryad, Figshare, Zenodo, OSF, or any similar platform.

The results (i.e. values for each ISO parameter for each surface) should also be included as SI.

9) Lines 289: do the authors mean λ_c (cut-off between waviness and roughness) or λ_s

(cut-off between noise and roughness)? If indeed λ_c , why did the authors choose not to remove the noise? Finally, 25 μm seems to be a high value for λ_s (usually 2.5 μm), but very low for λ_c (usually 250 or 800 μm).

10) Lines 303-305: why use a different test here? I am not familiar with matched pairs t-tests, but I do not see why an F-test ANOVA between the guilds (on raw data or on PCA scores) would not be appropriate.

11) Line 324: which version of package 'strap'? Please update your R installation; a lot has happened since 2010. What about adding the R scripts to the SI? This would increase reproducibility.

Results and Discussion

Note that the Tables S3 and S5 are unreadable because of formatting issues (see comment #21 below), so some of my comments are based only on the graphs.

12) Lines 101-102: I believe it should be "Fig. S3" here.

13) Line 107: I disagree with the statement that "specimens tend to cluster together rather than being distributed broadly". Just looking at Rhamphorhynchus, Dorygnathus and Pterodactylus on Fig. 2 shows that individuals from these taxa (most of those with more than 1 individual) do not cluster closely; they are spread over at least half of PC1.

14) Lines 125-126: there is so much overlap between the dietary categories that the pterosaurs overlap with almost all groups.

15) Lines 128-130: the 'softer' invertebrate consumers extend as far on PC1 as piscivores and carnivores, so this dietary category cannot be excluded for Istiodactylus.

16) Lines 137-139: Dimorphodon also overlaps with 'softer' invertebrate consumers.

17) Lines 147-151: Coloborhynchus does not overlap at all with piscivores.

18) Lines 152-165: What about taphonomy and duration of deposition? Could it be that they were not truly sympatric? A very brief discussion about sympatry is warranted here.

19) Lines 166-174: please add a new plot showing the ontogeny vs. DMTA. The data are completely missing (see comment #6 & 23).

SI

20) Fig. S1:

- This figure has already been published in Bestwick et al. 2019. Please clarify.
- This classification is valid only if there are no herbivores, which might not always be true.
- A 'hard' or 'soft' invertebrate feeder can still feed on up to 50% vertebrates. The thresholds are debatable.
- I suggest reorganizing the boxes for the end (with 'carnivore' at the end of the row and 'piscivore' just below), so that the organization is consistent.

21) Serious formatting issues with Fig. S6, Tables S3 and S5 render them unreadable (same problem in the original Word-file, although I managed to have a look at Fig. S6 there).

22) Table S4: Sds and Ssc are not part of ISO 25178-2 (2012). To which norm do they belong then?

23) SI text:

- Lines 67-69: Istiodactylus falls outside of the carnivore group but still within the piscivore group. So why assign it a carnivorous diet?
- Lines 69-70: Lonchodectes could as well be a 'harder' invertebrate consumer.
- Lines 72-73: Lonchodracho falls inside the 'harder' invertebrate, and outside of the carnivore, groups!
- Lines 78-79: Darwinopterus is far away from piscivores and carnivores!
- Lines 85-86: Rhamphorhynchus has a very broad diet, not just piscivory!
- Lines 89-91: Germanodactylus and Scaphognathus fall only in the 'hard' invertebrate range.
- Lines 92-93: show the data (see comment #6 & 19).
- Lines 111-122: this part deserves to be in the main text.

I tried to provide helpful and constructive comments; hopefully, they will indeed be.
Ivan Calandra

Reviewer #2:

Remarks to the Author:

Your work compiles a huge amount of data, which are well collected and documented with adequate methods. Because of the analytical load the palaeobiological impact of your paper is camouflaged but clearly should be better elaborated. If the points of discussion proposed in my attachment would become part of the discussion the paper would be a high impact one, because it would combine palaeobiological aspect with tough statistical data, which could be presented in a moer condensed way (there are some redundancies).

Good luck

Review Bestwick et al.

The paper provides a huge amount of Data concerning microwear of non-occlusive dentitions with respect to its implication on diet. Microwear pattern of bat canines and teeth of a selection of extant reptiles and pterosaurs are compared with each other in order to calculate the respective diet spaces and draw conclusions (1) on the diet guilds of a selected amount of pterosaurs, (2) the diet of a hypothetical basal-most pterosaur, niche partitioning or concurrence (example pterosaurs from the Frankonian Alb), (2) evolutionary trends within Pterosauria. The study is clearly innovative and ambitious and sure may yield a helpful tool for further studies, but mainly to that scientific community that deals with microwear, and this group is large, especially in the field of occlusive dentitions, namely advanced Synapsida. To what I can judge, the methods applied are appropriate, but I am not a specialist in the field of statistics. The conclusions based on original data retrieved from the taxa selected. Before I present my recommendation, please allow me to raise some issues.

Dietary guild assignments

It is not clear to me, on what basis the extant animals are grouped into their guilds. It is just cited that faeces and intestinal contents were used. There is a great amount on diet of crocodylians. I checked the literature cited and found that the diet not only depends on the size of the animals but also on the habitat, the season and the coexistence of two species in one and the same habitat. Crocodylians that inhabit lakes consume fish because this food source is abundant. In other places the prey items change with the availability. During the beginning of the rainy season the Nile crocodile feast almost exclusively on wildbeasts that cross the Mara River. With the exception of *Gavialis*, *Tomistoma*, and *C. cataphractus*, crocodylians eat what they can get. Thus it appears difficult to me how to assign them to guild per se without referring to the above mentioned parameters, the construction of the snout, the differentiation of the dentition and the adult size. There is also a bias on the size of animal that could be handled for a stomach flush. The same holds true for monitors, especially those with a wide range of distribution. Therefore it would be crucial to know the provenance of the samples specimens, because without this information only a statistical argument remains. If those data are not available these problems should at least be shortly addressed and should be the anatomical and size bias.

The hardness problem

Animals that can consume hard-shelled animals also can consume soft prey items that probably would not yield any trace on the tooth crown. The depth of a wear trace also depends on the hardness and thickness of the enamel. Not only that: the tooth inclination influences the direction of the wear traces, too, especially in non-occlusive dentitions, where especially inclined teeth of upper and lower jaw may cause wear facets against each other. Could the authors explicitly exclude such artefacts and briefly explain how?

Tooth anchoring

While crocodylians have a thecodont dentition with firmly implanted teeth, monitors are pleurodont and thus a more loose insertion. Additionally monitors have a highly amphikinetic skull. While forcing prey down, bite force vectors change from vertical to almost horizontal. This certainly influences the microwear and should thus be discussed.

Literature:

Riepel, O. R. 1979. A functional interpretation of the Varanid dentition (Reptilia, Lacertilia, Varanidae). *Gegenbaurs morphologisches Jahrbuch* 125(6): 797-817

McCurry MR, Mahony M, Clausen PD, Quayle MR, Walmsley CW, Jessop TS, et al. (2015) The Relationship between Cranial Structure, Biomechanical Performance and Ecological Diversity in Varanoid Lizards. *PLoS ONE* 10(6): e0130625. <https://doi.org/10.1371/journal.pone.0130625>

The bat analogue

It is well true that bats are the only actively flying dentulous amniotes in the Recent. Using the canines as an equivalent for a non-occlusive tooth is a perfect idea. The only problem is that bats are diphyodont, reptiles are polyphodont. This means that the canine of a bat is longer exposed to stress and worn during a bat's life. Did the authors have in mind that this could influence the microwear, too? Another thing is that in bats the diet is correlated with the flight apparatus, especially with respect to the jaw muscles. If this could hold true for pterosaurs as well, that would be a point of discussion with respect to pterosaurian evolution and change of diet.

Dumont, Elizabeth R. 2007 Feeding mechanisms in bats: variation within the constraints of flight. *Integrative and Comparative Biology*; DO 10.1093/icb/icm007.

Aguirre, L.F., Herrel, A., Van Damme, R. & Matthyssen, E. 2003. The implications of food hardness for diet in bats. *Funct. Ecol.* 17: 201-212 and citations therein.

Images

Maybe this was my copy only, but with the exception of the bat canine microwear I saw no other images. I would be great to have at least one plate with examples to visualize the differences. If there are such plates I would love to see them. Additionally a simplified line drawing of the wear would be great. Should do in the supplement.

Final comment and to the authors

Your work compiles a huge amount of data, which are well collected and documented with adequate methods. Because of the analytical load the palaeobiological impact of your paper is camouflaged but clearly should be better elaborated. If the points of discussion proposed above would become part of the discussion the paper would be a high impact one, because it would combine palaeobiological aspect with tough statistical data, which could be presented in a moer condensed way (there are some redundancies).

Recommendation

Accept with major revision

Karlsruhe, 6th of February 2020

Chief curator, head of geology department and professor for palaeontology, zoology and geocology

Reviewer #3:

Remarks to the Author:

The manuscript attempts to discern the diets of extinct flying reptiles, Pterosaurs, using non-occlusal texture-based microwear analysis. The authors first construct an extant "database" of microwear textures in reptiles and bats. They then project the microwear texture data for the extinct species into the same PCA space as a means of inferring their diets. Additionally, they use ancestral character estimation (contMap in Phytools, which performs ACE as part of the mapping procedure) to infer the likely ancestral diets of pterosaurs. Their results agree in some ways with studies using tooth and skull morphology to infer Pterosaur diets and provide novel dietary information for species with controversial diets.

Overall, the study is well done and the manuscript is well written. As far as I can tell, from pulling up the Lu phylogeny, they have done a good job sampling a wide array of Pterosaurs from across the phylogeny.

I, however, have some analytical concerns that I hope the authors are able to easily address. Some of these concerns may stem from the fact that, as someone who works on tooth wear but not reptiles, I am not familiar with the preceding studies (i.e. those developing the microwear texture method for use in reptiles). Others, I think could be relatively easily addressed with a little more R code.

1) The authors do not provide justification for the numbers of species included in the extant microwear texture database. The number of reptile species is, for example, smaller than the number of Pterosaur species sampled. Similarly, the number of bat species sampled seems vanishingly small, given they are the most diverse clade of extant mammals (~2,500 species). I am not suggesting that the authors sample 2,500 bat species or every extant reptile. But there needs to be some justification of why these species were sampled and how the authors believe that such small numbers of species could possibly represent the diversity of microwear textures. The authors have included a relatively large number of individuals per species, for which I applaud them. I still wonder, however, whether including only a small number of species in each dietary guild is representative of typical patterns of tooth wear within the guild. For mammalian herbivores, for example, certain guilds show much higher variation in the numbers and types of wear features than others. Perhaps this is not the case for reptiles or for non-occlusal microwear in reptiles. But then, I would like to see the authors discuss this.

2) The authors infer diets for Pterosaurs by projecting them into the PCA space constructed using extant species with known diets. This is, of course, an accepted method, but simply involves researchers deciding on a dietary guild based on visual inspection of the placement of species in PCA space. A more robust way of inferring the diets for extinct species would be to use a discriminant function analysis. Firstly, a DFA would provide an estimate of how well the data can discriminate among the dietary guilds of the extant species (which answers one of my additional questions, since there appears to be a great deal of overlap in the PCA plots). Secondly, a DFA provides a function that can be used to assign diets to unknowns. Perhaps, the authors have not used DFA because of their low rates of species sampling but it is the gold standard for these types of palaeodietary studies.

3) The authors use contMap as their method of ancestral character estimation. This is a great way of visualizing reconstructed values on the tree but the authors do not report the probability of each dietary guild at the ancestral node. These can be pulled out of the contMap object, although other methods of ACE provide more information on the probabilities. In short, the authors should go beyond just reading the colour off the contMap plot.

4) One more theoretical note and, perhaps, relating to my unfamiliarity with preceding studies, the present manuscript lacks any discussion of why non-occlusal microwear would be preferred to occlusal

microwear. I infer that it is because you're removing the influence of tooth-on-tooth wear and, presumably, only analyzing the food-on-tooth wear. But this also requires that food (particularly food that I assume is relatively non-abrasive like flesh) produce wear when simply being moved around in the mouth or perhaps when pierced etc. I know there has been some debate in the mammal literature regarding whether certain food items are tough enough to wear enamel in and of themselves. Additionally, as far as I understand, the food is probably not spending a long amount of time in a Pterosaurs mouth (it is caught and swallowed fairly quickly, I assume). Is this enough time for food items to produce tooth wear? How thick is Pterosaur enamel and how likely is it for insects/fish/bones etc. to produce non-occlusal wear? Do the types of wear present reflect what one would expect, given the properties of the food items? I would like at least a few sentences addressing this topic.

My hope is that the authors have the answers to these questions in their back pockets and are able to address them easily. It is an interesting study! The points about the DFA and contMap should be easily addressed with a few more lines of code.

Reviewer Comments Responses

We thank the reviewers for their detailed reviews and many helpful suggestions to improve clarity. We have adopted most of them and a full response is presented below. Reviewer comments and issues are written in **red**, with our responses written in **black** and indented for visual clarity.

Reviewer 1 Comments

We thank the reviewer for their careful scrutiny of the ms and their close attention to technical details. Addressing their concerns has allowed us make important clarifications and correct minor errors. Our responses to the reviewer's points have in nearly all cases resulted in changes to the manuscript, as detailed in our point-by-point response below.

Remarks to the author

This manuscript presents the first application of DMTA to pterosaurs to reconstruct their debated diets. Although pterosaurs just represent a new application of a well-established method (DMTA), it still greatly expands the field of applications. To do so, the manuscript builds on databases of extant reptiles (already published) and bats (new). This is the novelty of this manuscript.

This study will therefore be of interest to the DMTA and extinct reptile communities. My review focuses on the content and soundness of the manuscript. I will leave it up to the editor to decide whether or not it has a sufficient potential impact for Nature Communications.

The manuscript is well-written and the argument easy to follow.

Beside some minor comments (see below for details), I have two important issues:

The authors discuss almost exclusively the results of PC1. They should take PC2 into account too. This would give a more accurate (and nuanced) presentation and interpretation of the results. As is, some interpretations seem to over-interpret the data. This is true for the main text and for SI text as well. See also comments #15-17 & 23 below.

We appreciate this suggestion from the reviewer and have modified the text to clarify the role of PC2 in dietary interpretations. Most of our discussions are focussed on PC1, the more informative axis. PC axis 2 represents a relatively low proportion of the microwear texture variation in the extant reptile and bat texture-dietary spaces (25.1% and 16.9%; Figs. 1 and S3, respectively). PC2 exhibits fewer and weaker correlations with diet in extant reptiles, and therefore provides fewer constraints on pterosaur diets based on their separation along this axis. We have made this point explicit in the main and supporting texts. Nonetheless, following the reviewer's suggestion we have included more discussion of pterosaur diets based on their PC2 values in the main and supporting texts to give more comprehensive interpretations of pterosaur diets. See below for responses to comments 15–17 and 23.

We have also included the results of several analyses along PC2 in the main and supplementary text, including; (i) whether *Rhamphorhynchus* and *Pterodactylus* occupy separate areas of texture-dietary space (testing for sympatric dietary partitioning) and (ii) whether *Rhamphorhynchus* PC2 values correlate with specimen size (testing for ontogenetic dietary partitioning). These tests have been done for both the extant reptile and bat texture-dietary spaces to increase the robustness and transparency of our analyses.

A lot of the raw data are missing. As is, the study is not reproducible. I believe this is fundamental for any publication, and especially for Nature Research / Springer Nature. See also comments #6, 8 & 11 below.

The reviewer makes an excellent point, and we have now included the raw microwear texture data for all extant reptiles, bats and pterosaurs are now included in Table S1. We have also included the lower jaw lengths of all extant and extinct taxa for size comparisons. These size data are especially important for our *Rhamphorhynchus* microwear correlations with ontogeny (see comments below pertaining to this). The PC1 and 2 values for all specimens within the extant reptile and bat texture-dietary spaces are included in Table S1 for full transparency and reproducibility.

The R code used to produce the phylo-texture-dietary spaces and the ancestral dietary state reconstructions are included in the R file "Pterosaur_dental_microwear" as part of the supporting information. To address one of the main concerns of reviewer 3, we also produced PC1 and 2 value estimates for all the ancestral nodes in each of the three evolutionary trees used in our analysis (see comment 3 by reviewer 3, below). These PC1 and 2 value estimates, as well as the variance and 95% confidence intervals of each node in all three phylogenies, are included in Table S8 for full transparency.

Main text Introduction

1) Line 50: the impact of the evolution of pterosaurs on emergence and radiation of birds would be of interest too.

We agree with the reviewer's comment and have updated this sentence to: "...and what impact the emergence and radiation of birds had on pterosaur dietary evolution, and vice versa".

2) Lines 66-68: please also cite Winkler et al. 2019 here.

Done

3) Lines 68-71: the data show that it works, but I am still wondering how DMTA can work if there is no prolonged contact between food and teeth. This is not addressed in Bestwick et al. 2019. Could

the authors elaborate here (or in the discussion)? I think this is a very important aspect of this application to reptiles.

The reviewers point out that our data show that reptile DMTA work is important, but what their comment highlights is a broader area of current debate in microwear studies – although decades of research provides overwhelming evidence that microwear represents a powerful dietary proxy, these precise mechanisms by which food generates microwear are unclear (Lucas et al. 2008; Schulz et al. 2013; van Casteran et al. 2020). Detailed discussion of this point is beyond the scope of this contribution as Bestwick et al. (2019) does address the strength of the relationship between microwear textures and diet in extant reptiles (and non-occlusal dentitions in general). Their discussion includes recognition that the dietary signals preserved within reptile microwear textures are more generalised (e.g. proportions of vertebrates and invertebrates) than the more subtle signals found in DMTA studies based on occlusal mammal dentitions (e.g. distinguishing grazing and browsing herbivory). These differences in specificity were discussed by Bestwick et al. (2019) as the result of fewer tooth-food interactions in reptiles and lower biomechanical forces acting on non-occlusal tooth surfaces during food consumption. Despite these differences, as the reviewer notes, significant and informative dietary signals were indeed found within the tooth microwear textures of modern reptiles, supporting our use of the data as a multivariate framework for reconstructing the unknown diets of extinct reptiles in the present study. We feel that the best approach is to direct readers to Bestwick et al. (2019) for discussion of this matter rather than repeating this background information given that the focus of this paper is on pterosaurs.

Bestwick J. *et al.* (2019) *Sci. Rep.* **9**, 11691.

Lucas P. *et al.* (2008) *BioEssays* **30**, 374–385.

Schulz E. *et al.* (2013) *PLoS ONE* **8**, e56167.

van Casteren, A. D. S. *et al.* (2020) *Sci. Rep.* **10**, 582.

Methods

4) Lines 209-255: it is not clear here what comes from Bestwick et al. 2019 and what is new. It should be emphasized. See also comment #20.

We have now made the “Specimen material” subsection of the methods more explicit about what data is newly sampled and what comes from Bestwick et al. (2019). We have also emphasised in the “Extant reptile and bat microwear frameworks” subsection of the main text which results have been previously reported in Bestwick et al. (2019). See below for our response to comment 20.

Bestwick J. *et al.* (2019) *Sci. Rep.* **9**, 11691.

5) Lines 256-257: sample size is missing. It should not be necessary to look in SI and count manually to get sample sizes of the important material. As a comparison, there are more details about the reptile dataset, even though it is already published.

This is a fair comment, and we have now included the sample sizes of all individual reptile and bat species in the “dietary guild assignments” subsection of the methods. We have also made clear distinctions between the sample sizes of each species and of each dietary guild, so as not to confuse readers, as the latter is the independent variable in our dietary analyses.

6) Lines 265-266: how good is the correlation of jaw length with size and age? This is critical before using it as a proxy for size/age. Additionally, how do the authors define "most immature individuals" and "late-stage juveniles and adults" (lines 169-170)? Finally, where are the data?

Analysis of large numbers of *Rhamphorhynchus* specimens indicates strong linear relationships and high levels of correlation between different components of the skeleton, including skull elements (Bennett 1995; Prondvai et al. 2012). The skull and jaws therefore exhibit isometric growth with respect to other anatomical structures. This linear scaling indicates that lower jaw length can be used as a proxy for overall body size. We have modified the “Sampling strategy” subsection of the methods to make this more explicit. In response to the reviewer’s comment we have updated the text that reports the ontogenetic dietary shifts in *Rhamphorhynchus* because there is no unequivocal determinant of the size at which this pterosaur became sexually and/or osteologically mature. We use “highly immature” “juvenile” and “adult” in relative, rather than absolute, terms: “Specimens that are most likely immature individuals based on their size and degree of skeletal ossification (Bennett 1995; Prondvai et al. 2012) exhibit positive PC1 values, indicating invertebrate-dominated diets. Specimens that most likely represent late-stage juveniles and adults exhibit decreasing PC1 values and thus increasingly piscivorous diets.” We have explained our use of these terms and that we follow the analysis of Bennett (1995) in the methods section.

Lower jaw sizes for all extant reptiles and bats, and for all pterosaurs (where preservation quality permitted representative measurements) are now included in Table S1 for full transparency.

Bennett S. C. (1995) *J. Palaeontol.* **69**, 569–580.

Prondvai E. et al. (2012) *PLoS ONE* **7**, e31392.

7) Line 279: although not critical for focus variation please report the NA value of the 100x objective.

We have not made a change in response to this comment. This is not provided as part of the specification sheets for our instrument, and as the reviewer states, this is not critical for data capture using focus variation, or for producing reproducible methods.

8) Lines 283-291: why not use (or at least adapt) Arman et al. (2016)'s template? For example, the thresholding/removing outliers steps seem necessary for the present analysis: judging from Figs 1 and S3 (and color scales), I would expect high spikes/outliers to be present. Analyzing without excluding these aberrant points is dangerous, especially for height parameters (Sz, Sv, Sp). The results (i.e. values for each ISO parameter for each surface) should be included as SI.

Addressing each point in turn:

Using the Arman et al. template: We have opted not to change our approach. The reviewer is correct in that high spikes and outliers would be expected, and not excluding them would risk significantly distorting our results. We explain in the “Surface texture data acquisition” subsection of the methods that we employ manual deletion rather than thresholding: “All 3D data files were processed using Alicona IFM software (version 2.1.2) to remove dirt particles from tooth surfaces and anomalous data points (spikes) by manual deletion.” We also doublecheck all texture files for anomalous points when they are imported into Surfstand before the application of a fifth-order robust polynomial and robust Gaussian filter. This procedure follows standard laboratory protocols (e.g. Purnell et al. 2013; 2017; Gill et al. 2014; Bestwick et al. 2019). The digital surface elevation models from Fig. 1 and Fig. S3 are scale-limited texture files that have been processed for anomalous data point removal and the models within each figure have been set to the same topographic scale for easier comparisons.

Bestwick J. *et al.* (2019) *Sci. Rep.* **9**, 11691.

Gill P. G. *et al.* (2014) *Nature* **512**, 303–305.

Purnell M. A. *et al.* (2013) *J. Zool.* **291**, 249–257.

Purnell M. A. *et al.* (2017) *Biosurf. Biotribol.* **3**, 184–195.

Including results as SI: Done. The raw texture data for all 22 parameters initially used in analysis for all extant reptiles and bats, and for all pterosaurs, are now included in Table S1.

9) Lines 289: do the authors mean lambda c (cut-off between waviness and roughness) or lambda s (cut-off between noise and roughness)? If indeed lambda c, why did the authors choose not to remove the noise? Finally, 25 µm seems to be a high value for lambda s (usually 2.5 µm), but very low for lambda c (usually 250 or 800 µm).

We thank the reviewer for the opportunity to check and clarify this point, but our text is correct. We followed our standard lab protocols for generating scale limited surfaces using SurfStand software developed by the metrology group in Huddersfield led by Liam Blunt,

who had significant input into development of the ISO parameters upon which our analysis is based. We employed a standard combination of operators and filters to produce scale limited surfaces, and the filter in question does indeed have a cut-off of 0.025 μm as reported (the software allows this to be selected from a drop down menu with limited options). Although desirable, there are as yet no internationally agreed approaches to the generation of scale limited surfaces. Our goal is to be explicit in our methods so that our analyses are repeatable. Our methods for generating scale limited surfaces here are the same as those used in, for example, Purnell et al. (2013; 2017), Gill et al. (2014), Zhang et al. (2017), and Bestwick et al. (2019).

Bestwick J. *et al.* (2019) *Sci. Rep.* **9**, 11691.

Gill P. G. *et al.* (2014) *Nature* **512**, 303–305.

Purnell M. A. *et al.* (2013) *J. Zool.* **291**, 249–257.

Purnell M. A. *et al.* (2017) *Biosurf. Biotribol.* **3**, 184–195.

Zhang H. *et al.* (2017) *Quatern. Int.* **445**, 60–70.

10) Lines 303-305: why use a different test here? I am not familiar with matched pairs t-tests, but I do not see why an F-test ANOVA between the guilds (on raw data or on PCA scores) would not be appropriate.

The suggested approach is a good one. In fact we already report the results of ANOVA with Tukey HSD pairwise testing on dietary guild PC1 and PC2 scores to determine whether guilds occupied different areas of multivariate space along these axes. The reptile guild results are reported in Bestwick et al. (2019) and we have made this more explicit in the “DMTA statistical analyses” subsection of the Methods to prevent any confusion “ANOVA with pairwise testing was used on the PC axis 1 and 2 values of dietary guilds to determine whether guilds occupy different areas of multivariate space along these axes (see Bestwick et al. ¹⁸ for results for reptiles”. ANOVA results on the PC1 and 2 values between the bat guilds are reported in the Supporting Text.

We mention that the matched-pairs t-tests were employed to independently test for subtle microtextural differences between dietary guilds. This is in addition, not an alternative, to the ANOVA tests. Our use of matched-pairs testing for the bat dietary guilds (results reported in the Supporting Text) is the same method of statistical testing employed to delimit the reptile dietary guilds in Bestwick et al. (2019).

Bestwick J. *et al.* (2019) *Sci. Rep.* **9**, 11691.

11) Line 324: which version of package 'strap'? Please update your R installation; a lot has happened since 2010. What about adding the R scripts to the SI? This would increase reproducibility.

The strap version used is 1.4. This is now mentioned in the “pterosaur dietary evolution” subsection of the methods.

The R code generated to produce the phylo-texture-dietary spaces and the ancestral dietary state reconstructions are now included in the R file “Pterosaur_dental_microwear” as part of the supplementary information for reproducibility.

Results and Discussion

Note that the Tables S3 and S5 are unreadable because of formatting issues (see comment #21 below), so some of my comments are based only on the graphs.

12) Lines 101-102: I believe it should be "Fig. S3" here.

Thanks for spotting this. It has been corrected.

13) Line 107: I disagree with the statement that "specimens tend to cluster together rather than being distributed broadly". Just looking at *Rhamphorhynchus*, *Dorygnathus* and *Pterodactylus* on Fig. 2 shows that individuals from these taxa do not cluster closely; they are spread over at least half of PC1.

This sentence has been updated to better articulate our intended meaning. It now reads “...where pterosaur taxa are represented by multiple specimens, there is tendency for them to cluster together rather than exhibit a random distribution across the entire texture-dietary space (e.g. *Darwinopterus*, *Dimorphodon*) although some are more broadly distributed (e.g. *Dorygnathus*; *Pterodactylus*; Fig. 2; Fig. S2; Supplementary Text). This non-random distribution is noteworthy, given that pterosaur data played no role in structuring the PCA”

14) Lines 125-126: there is so much overlap between the dietary categories that the pterosaurs overlap with almost all groups.

We have added a sentence to clarify that although *Rhamphorhynchus* specimens overlap with other dietary guilds in the texture-dietary space, the distribution of *Rhamphorhynchus* corresponds more closely to piscivores than to any other group.

15) Lines 128-130: the 'softer' invertebrate consumers extend as far on PC1 as piscivores and carnivores, so this dietary category cannot be excluded for *Istiodactylus*.

We have updated this sentence for clarity. It now reads “Our DMTA results place *Istiodactylus* at the extreme end of the vertebrate scale with respect to reptile-based PC 1, plotting closest to a number of reptile carnivore samples (Fig. 2).” We explicitly mention that our interpretation of *Istiodactylus* as a carnivore is based both on its position within the

reptile texture-dietary space and with other independent lines of evidence (e.g. Witton 2012). We feel that this is sufficient justification for the dietary assignment of this pterosaur.

Witton, M. P. (2012) *PLoS ONE* **7**, e33170.

16) Lines 137-139: *Dimorphodon* also overlaps with 'softer' invertebrate consumers.

We have updated the sentence for increased clarity. It now reads: "The high degree of overlap of *Dimorphodon* tooth textures with reptile carnivores and piscivores across PCs 1 and 2 (Fig. 2) suggests that the diet of *Dimorphodon* largely consisted of vertebrates, although some consumption of 'softer' invertebrates cannot be entirely ruled out."

17) Lines 147-151: *Coloborhynchus* does not overlap at all with piscivores.

The text has been updated for clarity. The reviewer is partially correct in that *Coloborhynchus* and extant piscivores are separated along PC2 as the pterosaur has a more positive PC2 value. However, *Coloborhynchus* has a PC1 value of -0.422, which falls in the middle of the range of piscivores along this axis.

18) Lines 152-165: What about taphonomy and duration of deposition? Could it be that they were not truly sympatric? A very brief discussion about sympatry is warranted here.

This is an important consideration. *Pterodactylus* and *Rhamphorhynchus* are both represented by hundreds of specimens from the same geological localities within the Solnhofen Limestone sequences (see Barrett et al. 2008 and Witton 2013 for comprehensive syntheses on spatial and temporal pterosaur fossil distributions and Wellnhofer 1970; 1975, for data specific to the Altmühltal Formation and adjacent units). The data assembled by Wellnhofer (1970; 1975) demonstrate that these two taxa were sympatric.

We do, however, briefly discuss true sympatry of two other pterosaurs from Solnhofen, *Germanodactylus* and *Scaphognathus*, in our main and supporting texts. Our dataset comprises only one specimen from each genus, limiting the strength of our dietary interpretations based on their placements within the texture-dietary spaces. In addition, their fossils are found in much smaller numbers than *Pterodactylus* and *Rhamphorhynchus*. In addition, it has been suggested that *Germanodactylus* and *Scaphognathus* were not typical members of the Solnhofen community (e.g. Unwin 2006). Adding these caveats ensures that we do not overstate our conclusions on possible sympatric partitioning/competition for these pterosaurs.

Barrett P. M. et al. (2008) *Zitteliana* **28**, 61–107.

Unwin D. M. (2006) *Pi Press*. 352 pp.

Wellnhofer, P. 1970. *Bayerische Akademie der Wissenschaften, mathematisch-naturwissenschaftliche Klasse, Abhandlungen*, **141**, 1-133.

Wellnhofer, P. 1975. *Palaeontographica A*, **148**, 132-186.

Witton M. P. (2013) *Princeton Uni. Press*. 304 pp.

19) Lines 166-174: please add a new plot showing the ontogeny vs. DMTA. The data are completely missing.

This is an excellent point and we have taken the opportunity to add a main text figure that plots how diet (PC axis 1) changes with *Rhamphorhynchus* specimen size (new Fig. 3). The figure includes scaled skull diagrams of an example hatchling, juvenile and adult so that readers can easily see the size disparity between different life-history stages. The data upon which the figure is based are included in Table S1 for full transparency and reproducibility. We have also included a supplementary figure that plots how diet, as represented by the bat PCA, changes with ontogeny (see below). This is the new Fig. S4 and the PC1 data are included in Table S1. We thank the reviewer for this suggestion as we believe the new figures better demonstrate our arguments for ontogenetic dietary partitioning in *Rhamphorhynchus*.

Relationships between size in *Rhamphorhynchus* and PC2 values are not significant, but we report them in the main text for transparency and completeness and to further address the main comment of reviewer 1.

SI

20) Fig. S1:

• This figure has already been published in Bestwick et al. 2019. Please clarify.

The figure legend now specifies that the figure has been adapted from Bestwick et al. (2019) to include the guild classifications of the sampled bats in the present study. Retaining the guild classifications of the reptiles will save readers having to switch between two papers. Bestwick J. et al. (2019) *Sci. Rep.* **9**, 11691.

• This classification is valid only if there are no herbivores, which might not always be true.

We opted not to make any changes in this regard. The classification in Fig. S1 was created to provide a repeatable and transparent process by which to assign dietary guilds to the sampled extant reptiles and bats providing the multivariate framework for analysis of pterosaurs. In the present study, no extant herbivores were sampled and it could therefore be argued that it was superfluous to include this guild in Fig. S1. However, the classification system could easily be adapted in future studies to include herbivores by including an initial question along the lines of “Do plants comprise >80% of the diet”. Taxa that answer “yes” to this question would be assigned to the herbivore guild and taxa that answered “no” would proceed through the rest of the classification system outlined in Fig. S1.

- A 'hard' or 'soft' invertebrate feeder can still feed on up to 50% vertebrates. The thresholds are debatable.

The reviewer is correct in their point about invertebrate consumers having diets that can theoretically comprise up to 50% vertebrates. Our guild percentage thresholds are based in part on ecological studies that assigned taxa to guilds based on what makes up >50% of their diet by volume or frequency (Losos & Greene 1988; González et al. 2014; Bregman et al. 2016; Becker et al. 2018). As this system has been utilised for many taxon groups, including mammals and reptiles, these studies were a reliable starting point for us. We adjusted some of the classification thresholds based on experimental studies that quantified the relative hardness of prey items as food (Evans & Sansom 2005; Kaliontzopoulou et al. 2012; Runemark et al. 2015; Dollion et al. 2017). This is because we are more concerned about differences in the material properties of food items, rather than the taxonomic compositions of prey. More experimental work has been done on the exoskeletons of invertebrates, which possess chitin and/or calcium carbonate that is difficult to fracture (Bestwick et al. 2019; and references therein). It therefore stands to reason that invertebrates will have disproportionate influences on microwear formation. Our classification system for choosing the percentage thresholds (as seen in Fig. S1), and assigning our extant taxa to these guilds are transparent.

Becker D. J. *et al.* (2018) *Environ. Poll.* **233**, 1076–1085.

Bestwick J. *et al.* (2019) *Sci Rep.* **9**, 11691.

Bregman T. P. *et al.* (2016) *Proc. R. Soc. B.* **283**, 20161289.

Dollion A. Y. *et al.* (2017) *Funct. Ecol.* **96**, 671–684

Evans A. R. & Sansom G. D. (2005) *Aust. J. Zool.* **53**, 9–19.

González C. *et al.* (2014) *Rev. Mex. Biodivers.* **85**, 931–941.

Kaliontzopoulou A. *et al.* (2012) *Ecol. Evol.* **26**, 825–845.

Losos J. B. & Greene H. W. (1988) *Bio. J. Linn. Soc.* **35**, 379–409.

Runemark A. *et al.* (2015) *Ecology* **96**, 2077–2092.

- I suggest reorganizing the boxes for the end (with 'carnivore' at the end of the row and 'piscivore' just below), so that the organization is consistent.

Done.

Other SI

21) Serious formatting issues with Fig. S6, Tables S3 and S5 render them unreadable (same problem in the original Word-file, although I managed to have a look at Fig. S6 there).

We apologise for the reviewer's difficulty in reading the supporting information. We have moved Tables S3 and S5 into Excel files so they are easier to read. We have re-saved all submitted documents according to the journal's guidelines so that editors and reviewers can easily look at the figures again should they wish.

22) Table S4: Sds and Ssc are not part of ISO 25178-2 (2012). To which norm do they belong then?

The reviewer is correct, and we are grateful that he pointed out this oversight. The surfstand software we use, in addition to ISO 25178-2 parameters generates two that predate it: Sds and Ssc. This is now explicit in Table S4.

23) SI text:

The reviewer provides detailed comments on the distribution of pterosaur data in the multivariate texture-diet space defined on the basis of bat diets. We are grateful for the opportunity to clarify our interpretations, but we note that we provide this analysis as an additional demonstration that non-occlusal microwear textures preserve dietary signals. Given the differences between bats, extant reptiles and pterosaurs, seeking consistency in the results regarding pterosaur diet of these independent analyses is a very stringent test of our approach, and a test that it passes. We address each specific comment below, and have also added this clarification to the text.

• Lines 67-69: *Istiodactylus* falls outside of the carnivore group but still within the piscivore group. So why assign it a carnivorous diet?

We have clarified our interpretation of *Istiodactylus* diet as the overlap of this pterosaur along PCs 1 and 2 in the bat texture-dietary space means that fish cannot be excluded from comprising part of the pterosaur's diet. However, the dietary correlations along PC 1 in the bat texture-dietary space taken together with the position of *Istiodactylus* in the reptile texture-dietary space supports the conclusion we draw in the main text: *Istiodactylus* was an obligate vertebrate consumer, most likely a carnivore". As we note, this accords with previous interpretations of diet based on comparative anatomy (e.g. Witton 2012).
Witton, M. P. (2012) *PLoS ONE* 7, e33170.

• Lines 69-70: *Lonchodectes* could as well be a 'harder' invertebrate consumer.

We have added an additional sentence stating that overlap with bat 'harder' invertebrate consumers along PCs 1 and 2 suggests that *Serradraco* may have consumed some 'harder' invertebrates. The name change from *Lonchodectes* to *Serradraco* reflects the latest revision of the taxonomic identity of the sampled specimen.

• Lines 72-73: *Lonchodracho* falls inside the 'harder' invertebrate, and outside of the carnivores!

This has been clarified, now reads “*Lonchodraco* exhibits a slightly more positive PC 1 value than *Serradraco*, falling within the ranges of bat invertebrate consumers (for PC2 also) and carnivores (Fig. S3e).” The name change from *Lonchodectes* to *Serradraco* reflects the latest revision of the taxonomic identity of the sampled specimen.

- **Lines 78-79: Darwinopterus is far away from piscivores and carnivores!**

We have clarified the wording, which, as the reviewer highlights, was previously misleading. “The PC1 and PC 2 values for *Darwinopterus* are most similar to bats that consume ‘hard’ invertebrate (Fig. 3e). This suggests that *Darwinopterus* was an almost exclusive invertebrate consumer, and while this is consistent with its position in the reptile dietary-texture space, it also overlaps with carnivores and piscivores in the latter analysis (Fig. 2). Some consumption of vertebrates cannot be ruled out on the basis of our DMTA.”

- **Lines 85-86: Rhamphorhynchus has a very broad diet, not just piscivory!**

We have clarified these sentences: “*Rhamphorhynchus* and *Pterodactylus*, as they do in the reptile analysis, exhibit significant separation in their distribution with respect to PC 1 ($t = -4.179$, d.f. = 18, $P = 0.0006$), but not PC 2 ($t = 1.286$, d.f. = 18, $P = 0.2148$; Fig. S3e). *Rhamphorhynchus* tends towards more negative PC 1 values, indicative of consumption higher proportions of vertebrates; specimens that are likely to be adults (based on their size and degree of skeletal ossification⁸) overlap most strongly with piscivores along PCs 1 and 2 (Fig. S3e).

- **Lines 89-91: Germanodactylus and Scaphognathus fall only in the 'hard' invertebrate range.**

While these pterosaurs do overlap with ‘harder’ invertebrate consumers along both PCs 1 and 2, they also exhibit PC1 values that fall within the ranges of ‘softest’ invertebrate consumers and, to a lesser extent, carnivores. This suggests diets that consist largely, but not wholly, of invertebrates. We have clarified our interpretations of *Germanodactylus* and *Scaphognathus* diet from the bat texture-dietary space and have made sure not to overstate our conclusions based on the available data

- **Lines 92-93: show the data (see comment #6 & 19).**

We have clarified this through addition of a supplementary figure that shows how the diet of *Rhamphorhynchus* changes with specimen size (see comment above concerning a similar query relating to the reptile based analysis). This is the new Fig. S4 and all figure numbering has been updated accordingly. We thank the reviewer for this suggestion as this figure very nicely showcases and strengthens our conclusions of ontogenetic dietary partitioning in

Rhamphorhynchus. The jaw length and PC1 and 2 data are included in Table S1 for full transparency and reproducibility.

- Lines 111-122: this part deserves to be in the main text.

We do not entirely agree with the referee on this point and would argue that describing the differences between the phylogenies is more appropriate for the supplementary text as these descriptions are not directly relevant to diet. These descriptions have also been greatly lengthened by the inclusion of a third phylogeny in our analyses (Wang et al. 2017), and by more comprehensive discussions of each phylogeny based on the ancestral state reconstructions of each node within the three phylogenies (see comment 3 by reviewer 3 below). We provide justification for the inclusion of the Lü et al. (2016) phylogeny in the main text in that it exhibits the highest stratigraphic congruence of the three phylogeny (Andres 2015). It could therefore be argued that the dietary evolution reconstructions, and the inferred implications resulting from them, are the most representative of the three. We believe that the current structure creates a clearer narrative of dietary evolution for the reader.

Andres, B. (2015) *5th International Symposium on Pterosaurs*. Uni. Portsmouth.

Lü, J. et al. (2016) *PLoS ONE* **11**, e0154888

Wang, X. et al. (2017) *Sci. Rep.* **7**, 42763.

I tried to provide helpful and constructive comments; hopefully, they will indeed be.

Ivan Calandra

Reviewer 2 Comments

We thank the reviewer for their detailed review, highlighting some important areas of ambiguity where our text lacked the clarity we desire. We have attempted to address these concerns through clarification and additional explanation in the manuscript currently under consideration. Almost all their comments have resulted in changes to the text, resulting in improved clarity and transparency, and in reduced ambiguity.

Main comments

Dietary guild assignments. It is not clear to me, on what basis the extant animals are grouped into their guilds. It is just cited that faeces and intestinal contents were used. There is a great amount on diet of crocodylians. I checked the literature cited and found that the diet not only depends on the size of the animals but also on the habitat, the season and the coexistence of two species in one and the same habitat. Crocodylians that inhabit lakes consume fish because this food source is abundant. In other places the prey items change with the availability. During the beginning of the rainy season the Nile crocodile feast almost exclusively on wildbeasts that cross the Mara River.

With the exception of *Gavialis*, *Tomistoma*, and *C. cataphractus*, crocodylians eat what they can get. Thus it appears difficult to me how to assign them to guild per se without referring to the above mentioned parameters, the construction of the snout, the differentiation of the dentition and the adult size. There is also a bias on the size of animal that could be handled for a stomach flush. The same holds true for monitors, especially those with a wide range of distribution. Therefore it would be crucial to know the provenance of the samples specimens, because without this information only a statistical argument remains. If those data are not available these problems should at least be shortly addressed and should be the anatomical and size bias.

The reviewer's general point about assignment to guilds is important and something to which we paid close attention in developing this analysis. We thank the reviewer for giving us an opportunity to provide further clarification. We have added several sentences in our methods (in the "dietary guild assignments" subsection) to explain how we chose the dietary studies upon which to base our guild assignments for each taxon. This includes meeting as many of the following criteria as possible: representative sample sizes, dietary compositions represented as volumetric data; and spatial proximity of the dietary study to the location(s) from which the museum specimens we sampled were collected. These text additions provide enhanced clarity. The dietary studies we use are cited in the main text and in Table S2 for full transparency.

We provide the provenance data (where known) for each sampled specimen in Table S1 to show our attempts to minimise effects of inter-population dietary differences.

Regarding crocodylians in particular, as the reviewer points out, some exhibit seasonal shifts in diet based on prey availability. For example, spectacled caimans consume more fish during the wet season and more invertebrates such as crabs during the dry season (Lavery & Dobson 2013). There are few studies of the rate of microwear formation in non-occlusal dentitions, shedding and replacement of teeth in crocodylians will reset microwear every few months (teeth are replaced every 3–6 months). Therefore, sampling specimens from specific times of the year can provide us with higher degrees of confidence that microwear will record what these animals were eating in the period prior to their deaths. With our spectacled caimans, for example, we sampled specimens that had died early in the dry season. Specimens should therefore still retain a "fish signal" in their microwear textures and were thus classified as piscivores in our study.

Many crocodylians show ontogenetic shifts in diet (e.g. Delany 1999, Platt et al. 2013). We now explicitly state that we sampled extant specimens as similar in size to each other as possible to minimise this confounding variable. We now include the lower jaw size (as a proxy for overall size) of all specimens for full transparency. Specimen availability allowed us to separate saltwater crocodiles into adults and juveniles which also exhibit ontogenetic-based dietary shifts (Sah & Stuebing 1996). We now explicitly mention this in our methods.

Ultimately, while we agree with the reviewer that perfect dietary data is hard to obtain (and probably doesn't exist), previous analyses (Bestwick et al. 2019; Winkler et al. 2019) have found significant relationships between diet and microwear texture in reptiles, and this is what is important in the context of the present analysis.

Bestwick J. *et al.* (2019) *Sci. Rep.* **9**, 11691.

Delany M. F. *et al.* (1999) *Proc. Ann. Conf. Southeast. Ass. Fish. Wildlife Ag.* **53**, 375–389.

Lavery D. M. & Dobson A. P. (2013) *Herpetologica* **69**, 91–101.

Platt S. G. *et al.* (2013) *J. Herpetol.* **47**, 1–10.

Sah S. A. M. & Stuebing R. B. (1996) *J. Trop. Ecol.* **12**, 651–662.

Winkler D. E. *et al.* (2019) *Proc. R. Soc. B.* **286**, 20190544.

The hardness problem

Animals that can consume hard-shelled animals also can consume soft prey items that probably would not yield any trace on the tooth crown. The depth of a wear trace also depends on the hardness and thickness of the enamel. Not only that: the tooth inclination influences the direction of the wear traces, too, especially in non-occlusive dentitions, where especially inclined teeth of upper and lower jaw may cause facets against each other. Could the authors explicitly exclude such artefacts and briefly explain how?

Addressing each of the above points in turn:

Consuming both hard and soft prey: Items that are harder to pierce will indeed have disproportionate effects on microwear texture formation. As such, soft items that may have formed parts of pterosaur diets may be “obscured” by harder items. Throughout the main and supporting texts, we have taken care not to over-interpret our data by referring to the possibility of softer items in the diets of pterosaurs that overlap with extant ‘harder’ invertebrate consumers.

Wear trace depth and enamel thickness: This comment is very similar to the final comment raised by reviewer 3 (Comment 4). Please see our response to both these comments below.

Direction of wear traces: The reviewer is correct to point this out, although our approach is not based on the identification of individual wear traces, or their orientation. ISO 25178 parameters capture different aspects of the texture of a surface, and those that were informative in our study are not linked to the directionality of features. Furthermore, quantifying directionality from non-occlusal dentitions has been shown to be uninformative in dietary reconstructions (Winkler et al. 2019). Directionality is more appropriate for understanding chewing behaviours from occlusal dentitions (i.e. teeth that regularly experience tooth-tooth contact; Ungar et al. 2003). Regarding facets, as the reviewer notes, these are occasionally found in extant and extinct reptiles that possess non-occlusal dentitions (Schubert & Ungar 2005; Ősi 2011). These facets are easily identifiable as they

are vertically orientated, elliptical in shape and contain parallel features (Schubert & Ungar 2005; Ósi 2011). A handful of pterosaur taxa have been described with these types of facets (Ósi 2011), but we were unable to access any for inclusion in this study. We have added a sentence in the “Sampling strategy” subsection of our methods to explicitly state that we did not sample microwear from any facets that may have been the result of repeated tooth-tooth contact. This is an important distinction and should now eliminate any ambiguity on this topic.

Ósi A. (2011) *Lethaia* **44**, 136–152.

Schubert B. W. & Ungar P. S. (2005) *Acta. Palaeontol. Pol.* **50**, 93–99.

Ungar, P. S. *et al.* (2003) *Scanning* **25**, 185–193.

Winkler D. E. *et al.* (2019) *Proc. R. Soc. B.* **286**, 20190544.

Tooth anchoring. While crocodylians have a thecodont dentition with firmly implanted teeth, monitors are pleurodont and thus a more loose insertion. Additionally monitors have a highly amphikinetic skull. While forcing prey down, bite force vectors change from vertical to almost horizontal. This certainly influences the microwear and should thus be discussed.

Literature:

Riepel, O. R. 1979. A functional interpretation of the Varanid dentition (Reptilia, Lacertilia, Varanidae). *Gegenbaurs morphologisches Jahrbuch* 125(6): 797-817

McCurry MR, Mahony M, Clausen PD, Quayle MR, Walmsley CW, Jessop TS, *et al.* (2015) The Relationship between Cranial Structure, Biomechanical Performance and Ecological Diversity in Varanoid Lizards. *PLoS ONE* 10(6): e0130625. <https://doi.org/10.1371/journal.pone.0130625>

The reviewer is correct that crocodylians and monitor lizards have different tooth implantation and skull structures. These differences were addressed and discussed as a potentially confounding variable in microwear formation in Bestwick *et al.* (2019), which was the first application of DMTA to reptile dietary guilds that included both archosaurs and lepidosaurs. That reptiles did not separate in the texture-dietary space according to tooth implantation or skull construction (e.g. one cluster of thecodont dentitions and one cluster of pleurodont dentitions) indicates that if these anatomical differences have any effect on microwear formation, it is less than the signal from diet. We have added to our introduction: “This relationship between microwear texture and diet in taxa that do not chew food items provides a robust multivariate framework for this study, allowing us to quantitatively test and constrain pterosaur dietary hypotheses...”. Since the present study focuses on pterosaurs and tests different hypotheses to the Bestwick *et al.* (2019) study we feel it would be superfluous to repeat this discussion.

Bestwick J. *et al.* (2019) *Sci. Rep.* **9**, 11691.

The bat analogue. It is well true that bats are the only actively flying dentulous amniotes in the Recent. Using the canines as an equivalent for a non-occlusive tooth is a perfect idea. The only problem is that bats are diphyodont, reptiles are polyphodont. This means that the canine of a bat is longer exposed to stress and worn during a bat's life. Did the authors have in mind that this could influence the microwear, too? Another thing is that in bats the diet is correlated with the flight apparatus, especially with respect to the jaw muscles. If this could hold true for pterosaurs as well, that would be a point of discussion with respect to pterosaurian evolution and change of diet.

We are glad that the reviewer agrees with our strategy of sampling bat canines for our study. We have added an extra sentence in the "sampling strategy" subsection of our methods to improve clarity, stating that "Canines were sampled over premolars and molars because they represent closer functional analogues to reptile teeth since they are not used for chewing".

Microwear textures accumulate on tooth surfaces over the previous few weeks to months prior to sampling or the last few weeks to months of an organisms' life (Teaford & Oyen 1989; Merceron et al. 2010; Schulz et al. 2013). Thus, there is a constant "rewriting" of microwear textures that records recently consumed items, rather than a continuous addition over time. The diphyodont vs polyphodont argument is therefore avoided.

The impacts of bat jaw musculature on bite force and diet have been studied (Santana et al. 2010; 2012; Santana 2016) but have focused on the same dozen or so species (out of ~1100), and the impacts of these studies on microwear formation are currently unknown. Jaw muscle reconstructions of bats and pterosaurs is an interesting avenue for future investigation, but is beyond the scope of this paper.

Merceron G. *et al.* (2010) *PLoS ONE* **5**, e9542

Santana S. E. (2016) *Funct. Ecol.* **30**, 557–565.

Santana S. E. *et al.* (2010) *Funct. Ecol.* **24**, 776–784.

Santana S. E. *et al.* (2012) *Evolution* **66**, 2587–2598.

Schulz E. *et al.* (2013) *PLoS ONE* **8**, e56167.

Teaford M. F. & Oyen O. J. (1989) *Am. J. Phys. Anthropol.* **80**, 447–460.

Images. Maybe this was my copy only, but with the exception of the bat canine microwear I saw no other images. I would be great to have at least one plate with examples to visualize the differences. If there are such plates I would love to see them. Additionally a simplified line drawing of the wear would be great. Should go in the supplement.

We apologise for reviewer's difficulty in seeing the supporting information. We have re-saved all the submitted documents according to the journal's guidelines so that editors and reviewers can look at the figures again should they wish.

The scale-limited surfaces of the tooth textures in Figs 1 and S3 already serve as example illustrations for texture comparisons between extant dietary guilds and extinct taxa in the texture-dietary spaces. A simplified line drawing is not possible; the differences in microwear captured by the complex suite of texture parameters cannot be reduced to lines on a page. Our analysis is not based on identifying and counting numbers of microwear features (e.g. scratches and pits) present from a 2D image.

Final comment and to the authors

Your work compiles a huge amount of data, which are well collected and documented with adequate methods. Because of the analytical load the palaeobiological impact of your paper is camouflaged but clearly should be better elaborated. If the points of discussion proposed in my attachment would become part of the discussion the paper would be a high impact one, because it would combine palaeobiological aspect with tough statistical data, which could be presented in a more condensed way (there are some redundancies).

Recommendation

Accept with major revision

Eberhard Frey, Karlsruhe 6th February 2020

Chief curator, head of geology department and professor for palaeontology, zoology and geocology

Main text comments

All spelling and grammatical typos highlighted in the PDF have been corrected. We thank the reviewer for identifying all these errors.

- **Line 29; Check the sentence**

Done. Sentence now reads "...as most rely on morphological functional analogies".

- **Line 33; Why in quotes?**

Our use of these terms, such as 'hard' and 'soft', in quotes follows previous analyses, from the Leicester group, of microwear in non-herbivorous taxa. 'Hard' and 'soft' refer to the ease with which prey is pierced and chewed. This is perhaps most accurately expressed using the concept of prey 'intractability' developed by Evans and Sansom (2005) but the term is not widely used or understood so we utilise 'hard' and 'soft' as a shorthand. This is briefly explained in the "dietary guild assignment" subsection of our methods with a citation to a more complete explanation in Bestwick et al (2019).

Bestwick J. *et al.* (2019) *Sci. Rep.* **9**, 11691.

Evans, A. R. & Sanson, G. D. (2005) *Aust. J. Zool.* **53**, 9–19.

- **Line 43; What would be a non-integral part?**

This is a good point. We have clarified the text to improve readability.

- **Line 49; Systematically, birds are reptiles, too.**

Sentence corrected, now reads "...more similar to those of extant birds or non-avian reptiles..."

- **Lines 88 and 89; Harder than what? Hard-shelled versus soft-shelled?**

See above comment concerning Line 33 for our response.

- **Line 89; Isn't that too generalised? The Mesozoic saw the replacement of ganoid by teleosts, and the surface properties between these two are dramatic.**

Ganoine is a mineralised structure similar to enamel only found in a few groups of extant fish such as gars (Chen et al. 2012). The scales of most other fish in contrast possess collagen (Chen et al. 2012). Experimental work has shown that ganoine is over three times harder than collagen (Chen et al. 2012). It could therefore be argued that diets with higher proportions of ganoine scales will result in rougher tooth textures. Fish taxa that possessed ganoine scales were more common and diverse in the Mesozoic (e.g. Ebert et al. 2015), which could be a confounding variable for pterosaur microwear analysis.

The fish that are consumed by our extant reptile and bat piscivores almost entirely possess collagen scales (Brooke 1994; Delany et al. 1999; Wallace & Leslie 2008; Laverty & Dobson 2013). We therefore cannot split piscivores into subgroups based on the proportions of scale types. In our analyses, both reptile and bat piscivores have the smoothest microwear textures. We identify several pterosaurs that overlap with both reptile and bat piscivores along PCs 1 and 2, including *Boreopterus*, *Serradraco* and *Rhamphorhynchus* adults (Fig. 2; Fig. S3). These pterosaurs have previously been hypothesised as piscivores based on qualitative lines of evidence such as comparative anatomy and fossilised stomach contents (Bestwick et al. 2018; and references therein). DMTA therefore corroborates these previous studies. If the fish they consumed had high proportions of ganoine, we may have expected them to have more positive PC1 and 2 values, associated with rougher textures. As this was not the case it is unlikely that ganoine had a confounding effect on our analyses.

Bestwick J. et al. (2018) *Biol. Rev.* **93**, 2021–2048.

Brooke A. P. (1994) *J. Mammal.* **75**, 212–218.

Chen P.-Y. et al. (2012) *J. Mat. Res.* **27**, 100–112.

Delany M. F. et al. (1999) *Proc. Ann. Conf. Southeast. Ass. Fish. Wildlife Ag.* **53**, 375–389

Ebert M. et al. (2015) *PLoS ONE* **10**, e0116140

Lavery T. M. & Dobson A. P. (2013) *Herpetologica* **69**, 91–101.

Wallace K. M. & Leslie A. J. (2008) *J. Herpet.* **42**, 361–368.

- **Line 134; Relative to what?**

Sentence corrected, now reads “In contrast to the few pterosaur taxa where dietary hypotheses are relatively uncontroversial, DMTA gives us new insights into pterosaurs that have poorly constrained diets”.

- **Line 141; Check the grammar**

Sentence corrected and split into two for clarity, now reads “*Serradraco* is located towards the extreme end of the reptile carnivore/piscivore scale and overlaps more strongly with reptile piscivores long PC 2 (Fig. 2). This suggests that *Serradraco* was predominantly piscivorous.” This correction also explicitly uses the PC 2 data of this pterosaur to constrain its diet, addressing one of the main comments of reviewer 1 (see above). The name change from *Lonchodectes* to *Serradraco* reflects the latest revision of the taxonomic identity of the sampled specimen.

- **Line 236; These are hard!**

We based our hardness classifications on experimental studies that quantified forces needed to penetrate invertebrate exoskeletons (Kaliontzopoulou et al. 2012; Runemark et al. 2015; Dollion et al. 2017). Forces needed to penetrate ant exoskeletons were intermediate to those of beetles (the highest forces needed) and butterflies (lowest forces needed). We therefore assigned ants to our intermediate category; “softer” invertebrates. We have made this more explicit in the “dietary guild assignments” subsection of our methods.

Dollion A. Y. et al. (2017) *Funct. Ecol.* **96**, 671–684

Kaliontzopoulou A. et al. (2012) *Ecol. Evol.* **26**, 825–845.

Runemark A. et al. (2015) *Ecology* **96**, 2077–2092.

- **Line 237; Diplopoda have a calcified shell. And chilopoda would range in softer.**

Our hardness classifications of diplopods and chilopods are based on the same experimental studies mentioned in the previous comment.

- **Lines 238 & 242; Why in quotes?**

We thank the reviewer for pointing this out. The quotations have been removed from the terms juvenile and adult to reduce ambiguity.

- **Lines 238–255; What is the base for these assignments? Are these referred to diet analyses? Most crocs are generalists that eat what they can take down. Even herbivory is frequently**

observed. The same holds true for many monitors. The prey of crocs and monitors is often contaminated with sand, which is abrasive. What is the influence of permanent tooth replacement on microwear when compared with the dentition of bats where there is only one.

Addressing each of these points in turn:

What is the base for these assignments? Are these referred to diet analyses?

As we note above in our response to the first and fourth item of this reviewer's "main comments", assignment to guilds is important and something to which we paid close attention in developing this analysis. We thank the reviewer for allowing us to clarify; for details please refer to our response to the first and fourth item of reviewer 2's "main comments". We have included more information in the "dietary guild assignments" subsection of the methods detailing how we chose the studies used to assign extant taxa to guilds. These studies are highlighted in Table S2 and our guild classification system is in Fig. S1 for full transparency.

Most crocs are generalists... same holds true for many monitors. Our dietary guilds are treated as discreet/fixed variables for ANOVA and matched-pairs statistical testing e.g. the alligator and gharial are both assigned as piscivores, to test the null hypothesis that microwear textures do not vary between guilds. To test the subsequent null hypothesis that texture differences are not due to diet, we then treated dietary characteristics as continuous variables (e.g. proportion of diet that comprises fish, acquired from the dietary studies cited and in Table S2) for Spearman Rank correlations between diet and PC scores. For example, there is a difference between alligators and gharials in the degree to which they are piscivorous (fish comprise 57.5% of alligator diets and comprise 90% of gharial diets (Table S2)). This particular approach allows us to take dietary variation and generalism within and between guilds into account.

The prey of crocs and monitors is often contaminated with sand, which is abrasive. The importance of abrasives such as dust and grit on microwear formation is a very good point, but is a much disputed topic among microwear researchers. There are some claims that it is more important than dietary differences (e.g. Sansom et al. 2007; Lucas et al. 2013), while others argue that diet is more important (e.g. Merceron et al. 2016). Or it could reflect interactions between the two (Hua et al. 2015). The role of abrasives in microwear formation in reptiles however, is unknown. However, that two independent studies have found statistically significant relationships between diet and microwear textures in unrelated reptiles (Bestwick et al. 2019; Winkler et al. 2019), provides robust evidence that the presence of consumed abrasives does not mask dietary signals. While it remains possible that extraneous variables may have some effect on microwear formation, these independent analyses show that tooth microwear textures in reptiles preserve a dietary signal.

What is the influence of permanent tooth replacement on microwear when compared with the dentition of bats?. Studies of microwear texture from non-occlusal tooth surfaces are relatively rare, and so far as we are aware there have not been any studies that directly compares non-occlusal textures from animals with similar diets, but differences in the permanence of their dentition. We avoided sampling newly erupted teeth in extant and extinct reptiles so that the teeth we sampled were likely to have had sufficient time to accumulate dietary signals on their surface textures. Ultimately, our approach was based on testing the hypothesis that a multivariate texture analysis based on bats would indicate similar dietary preferences for pterosaurs as the analysis based on reptiles. That the results are not identical may relate to the reviewer's concerns about replacement. However, in that the results generate a very similar outcome regarding pterosaur diets suggests that the possible confounding effects of tooth replacement patterns were low.

Bestwick J. *et al.* (2019) *Sci. Rep.* **9**, 11691.

Hua, L. C. *et al.* (2015) *Am. J. Phys. Anthropol.* **158**, 769–775.

Lucas P. W. *et al.* (2013) *J. R. Soc. Interface* **10**, 20120923.

Merceron G. *et al.* (2016) *Proc. R. Soc. B.* **283**, 20161032.

Sansom G. D. *et al.* (2007) *J. Archaeol. Soc.* **34**, 526–531.

Winkler D. E. *et al.* (2019) *Proc. R. Soc. B.* **286**, 20190544.

Supporting Info comments

- **What kind of invertebrates could a juvenile *Rhamphorhynchus* seize with its dentition?**

This is a good question. However, one of the main reasons we applied microwear texture analysis to pterosaurs was to generate robust quantitative evidence of diet rather than qualitative, more speculative, observations. Our approach does not enable us to infer the taxonomic identities of food items consumed, only the material properties.

- **There are remains of large *Pterodactylus* that suggest that most of the specimens are juveniles.**

We mention in the “Pterosaur dietary evolution” subsection of the methods that we selected the largest individual from each genus for the dietary evolution reconstructions. In the case of *Pterodactylus* this included two specimens that are among the largest individuals assigned to this genus and both of which exhibit evidence (e.g. fusion of composite elements such as the scapulocoracoid) that indicate osteological maturity. The three remaining specimens are only approximately half the size of the two largest and show evidence of immaturity. Consequently, this broad sample for *Pterodactylus*, including adults, did not affect the results of, or our interpretations of, the dietary evolution reconstructions.

- **Check grammar**

Done. Sentence now reads "...do not evolve independently the other pterosaurs in the Andres & Myers phylogeny..."

- Of the crocs ranked within piscivores, only *Gavialis* is matching. All other also eat tetrapods of all kind and act as opportunists. Komodo dragons and some of the other monitors scavenge, eat large insects and whatever they get. Is there any literature on the extant examples with reference to the range and amount of prey?

This comment is very similar to the first of the "main comments" made by this reviewer, and we refer to our response to that comment to avoid repetition.

- Where can I find the respective literature?

The Fig. S1 legend has been updated and now directs readers to Table S2 for the proportional breakdowns of reptile and bat diets and the studies from which they are derived.

- Figure S2. Well, the overlap area is enormous

It is unclear whether the reviewer is suggesting a change here, or not. The comment relates to a plot that depicts the same analysis (with the addition of sample numbers) presented in the main text (Figure 2). Our interpretations of the results do not rely solely on overlap of convex hulls, so we have made no changes.

- Figure S3. One point is that those that can eat the hard stuff can also consume soft stuff, but not vice versa. Is there any correlation between diet and hardness of the enamel.

As far as we are aware, there is no published research on the relationship between enamel hardness and diet in bats. Purnell et al. (2013) demonstrate a relationship between diet and microwear texture in bats that successfully detects dietary differences between close relatives (cryptic subspecies of the same species) based on microwear texture. It is thus extremely unlikely that the differences in texture reflect differences in enamel hardness. We have made efforts in the main and supporting texts to ensure that when interpreting the diets of pterosaurs that overlap with consumers of harder invertebrates in the texture-dietary spaces that we do not automatically rule out softer invertebrates from forming at least part of the diet, as the former will have disproportionate effects on microwear formation.

Purnell M. A. et al. (2013) *J. Zool.* **291**, 249–257.

- Figure S5. At least *Boreopterus* could be a filter-feeder, probably *Coloborhynchus* as well for larger items.

The reviewer is correct in that *Boreopterus* and *Coloborhynchus* may have filter-fed on small aquatic organisms in addition to fish consumption (Bestwick et al. 2018; and references therein). However, our analyses cannot explicitly test this hypothesis as none of

the extant reptiles or bats used to generate the multivariate space upon which our analysis is based exhibit filter-feeding behaviour. We chose not to speculate about questions our analysis cannot explicitly address.

Bestwick J. *et al.* (2018) *Biol. Rev.* **93**, 2021–2048.

Reviewer 3 Comments

Remarks to the Author:

The manuscript attempts to discern the diets of extinct flying reptiles, Pterosaurs, using non-occlusal texture-based microwear analysis. The authors first construct an extant “database” of microwear textures in reptiles and bats. They then project the microwear texture data for the extinct species into the same PCA space as a means of inferring their diets. Additionally, they use ancestral character estimation (contMap in Phytools, which performs ACE as part of the mapping procedure) to infer the likely ancestral diets of pterosaurs. Their results agree in some ways with studies using tooth and skull morphology to infer Pterosaur diets and provide novel dietary information for species with controversial diets.

Overall, the study is well done and the manuscript is well written. As far as I can tell, from pulling up the Lu phylogeny, they have done a good job sampling a wide array of Pterosaurs from across the phylogeny.

I, however, have some analytical concerns that I hope the authors are able to easily address. Some of these concerns may stem from the fact that, as someone who works on tooth wear but not reptiles, I am not familiar with the preceding studies (i.e. those developing the microwear texture method for use in reptiles). Others, I think could be relatively easily addressed with a little more R code.

1) The authors do not provide justification for the numbers of species included in the extant microwear texture database. The number of reptile species is, for example, smaller than the number of Pterosaur species sampled. Similarly, the number of bat species sampled seems small, given they are the most diverse clade of extant mammals (~2500 species). I am not suggesting that the authors sample 2500 bat species or every extant reptile. But there needs to be some justification of why these species were sampled and how the authors believe that such small numbers of species could possibly represent the diversity of microwear textures. The authors have included a relatively large number of individuals per species, for which I applaud them. I still wonder, however, whether including only a small number of species in each dietary guild is representative of typical patterns of tooth wear within the guild. For mammalian herbivores, for example, certain guilds show much higher variation in the numbers and types of wear features than others. Perhaps this is not the case for reptiles or for non-occlusal microwear in reptiles. I would like to see the authors discuss this.

The reviewer is correct to note that we have not attempted to sample the diversity of extant taxa. Our experimental design was focussed on sampling taxa that would allow us to test hypotheses of the potential diets of pterosaurs, many of which were previously poorly constrained. Nonetheless, we have taken this opportunity to provide additional discussion of extant taxon choice, and of the factors that limited the availability of appropriate material, in the “dietary guild assignments” subsection of the methods. Factors included:

- The inappropriateness of some dietary guilds as pterosaur analogues. Using microwear textures from guilds that have ecological relevance to pterosaurs was more important than sampling the entire dietary diversity of extant clades. For example, many New World bats are nectarivores, sanguivores (blood feeders) or obligate frugivores (Santana et al. 2010), guilds that have never been suggested for dentulous pterosaurs (Bestwick et al. 2018; and references therein).
- Lack of robust dietary studies. We chose extant taxa that have diets that are well-constrained by stomach and/or faecal content studies giving volumetric breakdowns of dietary items (or frequency breakdowns as a minimum; Table S2). These breakdowns are crucial for testing the null hypothesis that microwear texture differences between dietary guilds are not caused by dietary differences. Taxa that lacked quantitative dietary data were excluded from the study.
- Museum sample sizes. Numbers of species, and of individuals from each species, was often limited by specimen availability. Dry skeletal specimens were favoured as the process of removing formaldehyde from tooth surfaces can weaken tooth enamel. Many reptile specimens (especially lepidosaurs) are often preserved in formaldehyde. Additionally, skeletal specimens may have missing or badly damaged teeth that prevents sampling. Adding these points to our methods section has increased transparency of our experimental design and increased the reproducibility of our sampling methods.

Bestwick J. *et al.* (2018) *Biol. Rev.* **93**, 2021–2048.

Santana S. E. *et al.* (2010) *Funct. Ecol.* **24**, 776–784.

2) The authors infer diets for Pterosaurs by projecting them into the PCA space constructed using extant species with known diets. This is, of course, an accepted method, but simply involves researchers deciding on a dietary guild based on visual inspection of the placement of species in PCA space. A more robust way of inferring the diets for extinct species would be to use a discriminant function analysis. Firstly, a DFA would provide an estimate of how well the data can discriminate among the dietary guilds of the extant species (which answers one of my additional questions, since there appears to be a great deal of overlap in the PCA plots). Secondly, a DFA provides a function that can be used to assign diets to unknowns. Perhaps, the authors have not

used DFA because of their low rates of species sampling but it is the gold standard for these types of palaeodietary studies.

We considered DFA (or linear discriminant analysis) and its potential use in the context of dietary analysis. Indeed, one of the authors has previous publications on microwear texture analysis that employ both PCA and DFA (e.g. Purnell *et al.* 2013; Purnell and Darras 2016). However, this experience led us to prefer for PCA, and this approach is consistent with other palaeodietary studies that combine DFA modern and extinct taxa (Delezene *et al.* 2013; 2016; Gill *et al.* 2014; Purnell *et al.* 2012; 2013; 2017; Purnell & Darras 2016; Bestwick *et al.* 2019; Winkler *et al.* 2019). This preference stems from the fundamental difference between PCA and DFA: PCA requires no a priori assumptions about dietary classes, whereas DFA does. This gives PCA an advantage over LDA for two reasons. First, the initial stage of our analysis was to determine whether microwear texture captures dietary signals in extant taxa for which diets are known. PCA, because it does not assume that taxa assigned to similar dietary classes should have similar microwear, does not impose a structure on the multivariate space that reflects the initial assumptions to the degree that DFA does. It allows for the possibility that taxa with similar diets do not have similar microwear, rather than 'trying' to allocate them to the same region of multivariate space. If the results of this phase of analysis reveal that there are significant relationships between diet and the principal axes that capture the highest proportion of the variation in texture, this can serve as a multivariate framework within which to test hypotheses about extinct taxa. Second, in the context of long extinct animals for which choices of precise dietary analogues among extant animals can be complex, this dietary framework approach is preferable to one that attempts to classify (such as DFA). There is no need to make any assumptions that the dietary classes of the extant animals capture the same dietary classes to which the extinct taxa belonged. Similarly we do not have to assume that the relationship between dietary class and microwear texture is precisely the same in distantly related extant and extinct taxa. It is important to reiterate here that in the way our study employs PCA the extinct taxa do not have any role in structuring the PCA space upon which their dietary interpretation is based. They are not constrained to plot according to any groupings reflected in the distribution of extant taxa, and in this respect our approach takes advantage of one of the most powerful aspects of PCA.

As we note above, a number of studies published in high quality peer reviewed journals, some of which are highly cited, employ the same PCA-based analytical approach that we have used here. While we agree with the reviewer that for some kinds of analysis DFA could be considered to represent a gold standard, we are of the opinion that PCA is more appropriate for our analysis, given the nature of our questions, uncertainties, and the data Bestwick J. *et al.* (2019) *Sci. Rep.* **9**, 11691.

Delezene L. K. *et al.* (2013) *J. Hum. Evol.* **65**, 282–293.

- Delezene L. K. *et al.* (2016) *Am. J. Phys. Antropol.* **161**, 6–25.
- Gill P. G. *et al.* (2014) *Nature* **512**, 303–305.
- Purnell M. A. and Darras L. P. G. (2016) *Surf. Topog.: Metrol. Prop.* **4**, 014006.
- Purnell M. A. *et al.* (2012) *J. R. Soc. Interface* **9**, 2225–2233.
- Purnell M. A. *et al.* (2013) *J. Zool.* **291**, 249–257.
- Purnell M. A. *et al.* (2017) *Biosurf. Biotribol.* **3**, 184–195.
- Winkler D. E. *et al.* (2019) *Proc. R. Soc. B.* **286**, 20190544.

3) The authors use contMap as their method of ancestral character estimation. This is a great way of visualizing reconstructed values on the tree but the authors do not report the probability of each dietary guild at the ancestral node. These can be pulled out of the contMap object, although other methods of ACE provide more information on the probabilities. In short, the authors should go beyond just reading the colour off the contMap plot.

We thank the author for this suggestion as we believe that the updated results of our dietary evolution reconstructions, and our interpretations from them, are now much more thorough and robust. We have produced ancestral PC1 and 2 value estimates for all major nodes of the evolutionary trees used in our analysis. We explicitly quote PC1 and 2 value estimates in the main and supporting texts for more robust statements on pterosaur dietary evolution (this also addresses one of the comments by reviewer 1). In addition to reviewer 3's suggestion, we performed ancestral dietary reconstructions on an additional evolutionary tree from Wang *et al.* (2017) to serve as an independent comparison (new Fig. S8) to the other two phylogenies and to generate more robust conclusions regarding pterosaur dietary evolution. The results using the Wang *et al.* (2017) phylogeny, and its implications for understanding pterosaur dietary evolution, are briefly mentioned in the main text and are elaborated on in the supporting text. We have also produced three new supporting figures (Figs. S9–11) which comprise non-time-calibrated evolutionary trees for the three phylogenies used in our analysis and each of which displays the position of the major ancestral nodes. Presenting the labels on separate figures avoids the ancestral character state figures appearing confusing. The PC1 and 2 value estimates, as well as the variance and 95% confidence intervals of each node in all three phylogenies, are included in Table S8 for full transparency.

Wang, X. *et al.* (2017) *Sci Rep.* **7**, 42763.

4) On a more theoretical note and, perhaps, relating to my unfamiliarity with preceding studies, the present manuscript lacks any discussion of why non-occlusal microwear would be preferred to occlusal microwear. I infer that it is because you're removing the influence of tooth-on-tooth wear and, presumably, only analyzing the food-on-tooth wear. This also requires that food (particularly food that I assume is relatively non-abrasive like flesh) produce wear when simply being moved

around in the mouth or perhaps when pierced etc. I know there has been some debate in the mammal literature regarding whether certain food items are tough enough to wear enamel in and of themselves. Additionally, as far as I understand, the food is probably not spending a long amount of time in a Pterosaurs mouth (it is caught and swallowed fairly quickly, I assume). Is this enough time for food items to produce tooth wear? How thick is Pterosaur enamel and how likely is it for insects/fish/bones etc. to produce non-occlusal wear? Do the types of wear present reflect what one would expect, given the properties of the food items? I would like at least a few sentences addressing this topic.

Addressing each main comment in turn:

Non-occlusal microwear: We agree with the reviewer. In the context of the dentitions analysed (reptiles and pterosaurs) “occlusion” is not the controlled, predictable relationship between two interlocking teeth, as it is in mammal molars, but is something that might occur in some individuals and not others within a species, sometimes fortuitously, and may also differ between taxa. In this context it is, as the reviewer notes, a confounding variable. Our sampling strategy, focused on food-on-tooth wear, was specifically designed to avoid this problem. Furthermore, the majority of toothed pterosaurs show no evidence of occlusal facets and the few specimens that do (Ősi 2011) were not sampled. As we note in our introduction, “Recent work has demonstrated that DMTA of non-occlusal (non-chewing) tooth surfaces of extant reptiles, including archosaurs (the clade to which pterosaurs belong), differs between dietary guilds^{18,20}. In addition, we have added some details to the “sampling strategy” subsection of the methods to be more explicit about our choice of non-occlusal surfaces.

Time spent consuming food: The reviewer touches on an interesting point here. To which the short answer is simple: we do not know precisely the mechanisms by which food creates microwear textures. The issue is surprisingly complex, and at present theoretical modelling and laboratory experiments are unable to fully explain the empirical relationship between microwear and diet that scores of studies have reported. The current study is not the first to apply DMTA to reptile non-occlusal tooth surfaces as work has been independently conducted by two research groups (Bestwick et al. 2019; Winkler et al. 2019). Taxa included multiple groups of modern lizards and crocodylians, which all exhibit minimal oral processing of food items (Cleuren & de Vree 2000; D’Amore et al. 2009). Both studies found dietary signals within reptile microwear (Bestwick et al. 2019; Winkler et al. 2019). While the preserved signals were broader than signals preserved within the occlusal microwear of extant mammals, these studies nevertheless show that signals can be preserved in non-occlusal dentitions. Furthermore, as mentioned in the introduction to our study, these studies show that they can serve as frameworks for reconstructing the diets of extinct reptiles that also did not chew. In addition, it is noted, in the main and supporting texts that the

placements of pterosaurs in the reptile and bat texture-dietary spaces are similar and that individuals of several taxa cluster closer together than they do to other taxa. As we note in our response to reviewer 2, previous analyses (Bestwick *et al.* 2019; Winkler *et al.* 2019) have found a significant relationship between diet and microwear texture in reptiles, and this is what is important in the context of the present analysis.

Pterosaur enamel: All microwear samples were obtained from enamel surfaces (i.e. none from surfaces where dentine was exposed), and the mechanical properties of enamel (e.g. resistance to external forces) are similar across a broad range of reptiles and mammals, despite the teeth of reptiles having much thinner enamel (Enax *et al.* 2013).

Regarding whether the wear present is what one would expect given the properties of food items, anything but general discussion is difficult given the lack of understanding of how the material properties of food are precisely related to the mechanism of microwear formation (see above). In a sense, however, this is the subject of the analysis presented here. Based on multivariate analysis of microwear texture (as captured through ISO texture parameters), our analysis explores how the microwear on the teeth of particular pterosaurs compares to that of extant taxa with known diets. Subsequently evaluating whether the microwear on pterosaur teeth is as we would expect results in a high degree of circularity in our reasoning. We have no way of independently evaluating this in detail.

The broader question regarding the relationship between food and microwear (e.g. does consumption of harder foods produce rougher surface textures?) can only be addressed through analysis of taxa with independent evidence of diet, and is the subject of the study of extant reptiles presented in Bestwick *et al.* (2019). We do not feel it would be appropriate to repeat the details of that analysis in the present contribution, but we have highlighted the broad relationship in the results, adding “In very broad terms, ‘harder’ foods correlate with rougher textures (see ref. 18 for details).”

Bestwick J. *et al.* (2019) *Sci. Rep.* **9**, 11691.

Cleuren J. & de Vree F. (2000) *Academic Press* 337–358 pp.

D’Amore D. C. *et al.* (2009) *Paleobiology* **35**, 525–552.

Enax J. *et al.* (2013) *J. Struct. Biol.* **184**, 155–163.

Ősi A. (2011) *Lethaia* **44**, 136–152.

Winkler D. E. *et al.* (2019) *Proc. R. Soc. B.* **286**, 20190544.

My hope is that the authors have the answers to these questions in their back pockets and are able to address them easily. It is an interesting study! The points about the DFA and contMap should be easily addressed with a few more lines of code.

Reviewers' Comments:

Reviewer #1:

Remarks to the Author:

The authors did a great job addressing my comments, as well as those of the other reviewers (as far as I can tell).

I just have two very minor comments:

- Regarding the R code, there is no "R file" in the submission. Only a TXT file that is an "Example R code".

I recommend saving the file in R format (.R) and providing more than just an example. The great advantage of scripts is that they can be shared. But it makes sense only if the scripts shared are the scripts that were used for the analysis.

- Regarding my original comment #3, my point was slightly different (I guess) so I would like to clarify:

I know that there is a lot of discussion and debate about how food generates microwear. But in all these discussions, it is assumed that food makes extensive contact with teeth.

In the case of reptiles, this contact is short and with much lower forces. So whatever particles are responsible for wear in general, the case of reptiles is still spectacular.

In other words, my point was that it is actually surprising that it even works! So my question was not "why it works less well than with e.g. mammals?" (this is indeed addressed in Bestwick et al. 2019) but rather "how can it even work?".

To me, the fact that it works implies that either (1) contacts are longer than expected, (2) forces are greater than expected, (3) there is more food-processing than expected, and/or (4) food itself does not directly produce microwear but maybe attrition is responsible (knowing that chewing behaviors are adjusted depending on the properties of the food, this could explain the relationship between DMTA and diet, i.e. correlation through a third factor).

But yes, maybe indeed beyond the scope.

Ivan Calandra

Reviewer #2:

Remarks to the Author:

Dear Authors,

subsuming all reviews and comments you added to the paper I have no problems in seeing that published. Concerning my comments I still have some problems but these concern science theory and biological approaches (I am biologist), which does not matter here. I understand that your major claim is to present a first DMTA analysis for pterosaurs based on extant analogues. The revised version is absolutely convincing and not only yields new testable models. I see no reason why not to publish the paper after having worked in the comments of all reviewers as I have read them. Congratulations.

Reviewer #3:

Remarks to the Author:

I reviewed a previous iteration of this manuscript. It was excellent to begin with and the authors have satisfactorily addressed my concerns. I therefore have no issues with the present manuscript being

accepted.

I now understand the justification for the species and specimens chosen for comparison. Thank you.

Point taken about the benefits of PCA over DFA.

I do think the results of the ancestral character estimations are clearer and more robust.

Thank you for the clarification regarding what is known and unknown regarding how non-occlusal wear is related to diet in non-mammals.

Congratulations on the excellent paper. I look forward to seeing it in press.

--

Danielle Fraser, PhD
Canadian Museum of Nature

Reviewer Comments Responses

We thank the reviewers for looking through our review commentary and for providing their final thoughts and suggestions. Our full response is presented below and we address the comments in the same order as given in the email and in the PDF of reviewer 2 (comments were provided using the sticky note tool in Adobe). Reviewer comments and issues are written in **red**, with our responses written in **black** and indented for visual clarity.

Reviewer 1

The authors did a great job addressing my comments, as well as those of the other reviewers (as far as I can tell).

I just have two very minor comments:

- Regarding the R code, there is no "R file" in the submission. Only a TXT file that is an "Example R code".

I recommend saving the file in R format (.R) and providing more than just an example. The great advantage of scripts is that they can be shared. But it makes sense only if the scripts shared are the scripts that were used for the analysis.

We have now saved the code as the R file "Supplementary Code 1" which is submitted with the final version of the manuscript within a ZIP file as asked by the editor. We used three separate phylogenies for our dietary evolution reconstructions and so our "example" is actually one of the three phylogenies we used. This can be easily substituted to use different phylogenies.

- Regarding my original comment #3, my point was slightly different (I guess) so I would like to clarify:

I know that there is a lot of discussion and debate about how food generates microwear. But in all these discussions, it is assumed that food makes extensive contact with teeth.

In the case of reptiles, this contact is short and with much lower forces. So whatever particles are responsible for wear in general, the case of reptiles is still spectacular.

In other words, my point was that it is actually surprising that it even works! So my question was not "why it works less well than with e.g. mammals?" (this is indeed addressed in Bestwick et al. 2019) but rather "how can it even work?".

To me, the fact that it works implies that either (1) contacts are longer than expected, (2) forces are greater than expected, (3) there is more food-processing than expected, and/or (4) food itself does not directly produce microwear but maybe attrition is responsible (knowing that chewing behaviors are adjusted depending on the properties of the food, this could explain the relationship between DMTA and diet, i.e. correlation through a third factor).

But yes, maybe indeed beyond the scope.

We thank the reviewer for clarifying this point. We agree that understanding the intrinsic and extrinsic factors behind the applicability of DMTA to reptiles is indeed a worthwhile avenue of further investigation and discussion. However, as the reviewer notes, this is perhaps best done in another study.

Ivan Calandra

Reviewer 2

Remarks to the Author

Dear Authors,

subsuming all reviews and comments you added to the paper I have no problems in seeing that published. Concerning my comments I still have some problems but these concern science theory and biological approaches (I am biologist), which does not matter here. I understand that your major claim is to present a first DMTA analysis for pterosaurs based on extant analogues. The revised version is absolutely convincing and not only yields new testable models. I see no reason why not to publish the paper after having worked in the comments of all reviewers as I have read them. Congratulations.

PDF comments

This is fine now, especially the revised table.

We are glad the reviewer approves of our change. No further updates needed.

They likely do.

We are glad the reviewer approves of our change. No further updates needed.

Perfect and important.

We are glad the reviewer approves of our change. No further updates needed.

Excellent.

We are glad the reviewer approves of our change. No further updates needed.

So you focussed on the hardness of prey? Just worth a mention.

We have taken care in our main and supporting texts to focus on the hardness of prey items, not their taxonomic identity, when interpreting pterosaur diets.

Ok, accepted

We are glad the reviewer approves of our change. No further updates needed.

Fine like this

We are glad the reviewer approves of our change. No further updates needed.

Very good now

We are glad the reviewer approves of our change. No further updates needed.

Ok

We are glad the reviewer approves of our change. No further updates needed.

I believe that for sure, but still cannot open the images but OK

We apologise for the reviewer's difficulty in seeing some of the supporting figures. These figures were saved according to the journal's guidelines and have now been resaved correctly with the editorial requests and comments taken on board.

As long as you explain these quote terms somewhere – fine with me.

These terms are now succinctly defined in the “dietary guild assignment” subsection of our methods to eliminate any confusion concerning the use of quotations.

Much better now!

We are glad the reviewer approves of our change. No further updates needed.

Fine

We are glad the reviewer approves of our change. No further updates needed.

See comment above

The terms ‘hard’ and ‘soft’ are now succinctly defined in the “dietary guild assignment” subsection of our methods to eliminate any confusion concerning the use of quotations.

I see your argument. Thanks for explaining, but when looking at the extremely pointed teeth of the tree guys and ganoine had no effect, one could conclude that the prey was seized with a minimum bite force with a kind of friction grip. An impact bite of elasmoid scales would certainly leave less traces on the tooth tip than an impact bite on a ganoid scale. The other option would be a rapid tooth replacement. Just thoughts.

These are very interesting points. With regards to pterosaurs using “minimum bite force”, while this feeding behaviour cannot be automatically ruled out, there is also no way of explicitly testing whether or not this is what happened. We have therefore chosen not to speculate this to keep our interpretations clear. With regards to tooth replacement rates, very little research has been done on this (e.g. Fastnacht 2008) and it is unclear if the same replacement rates can be reliably applied to all pterosaurs. Similarly, we have chosen not to speculate on this without more supporting evidence.

Fastnacht M. (2008) *J. Morphol.* 269, 332–348.

Fine

We are glad the reviewer approves of our change. No further updates needed.

Perfect. Fully understandable now.

We are glad the reviewer approves of our change. No further updates needed.

Well, I checked the paper and your comment is formally correct. Still, I have my problems with that, but this will not be a reason to reject.

The justification for assigning invertebrate prey into the distinct hardness categories are based on experimental studies and has been made explicit in the “dietary guild assignments” subsection of the methods.

Same comment as above. I had *Spirobolus* at home and *Scolopendra*. But again: formally correct.

The justification for assigning invertebrate prey into the distinct hardness categories are based on experimental studies and has been made explicit in the “dietary guild assignments” subsection of the methods.

This is much appreciated and accepted.

We are glad the reviewer approves of our change. No further updates needed.

Ok, I accept that argumentation. As a biologist I am more on the observe side.

We are glad the reviewer approves of our change. No further updates needed.

Well explained, thanks.

We are glad the reviewer approves of our change. No further updates needed.

Did you include this critical remark?

We have included the following sentences in Supplementary Note 1: “These results are broadly similar to microwear texture differences between vertebrate and invertebrate consuming guilds of extant reptiles. Reasons for why they are not the same are unclear but may reflect different tooth replacement patterns between reptiles and mammals. Further investigation into this, however, is beyond the scope of this study.”

Ok. Just an idea: there are numerous arthropods known from the laminated limestone of the Franconian Alb that could give you a qualitative support.

We are glad the reviewer approves of our response. No further updates needed.

Ok. The ontogeny argument is fine. I saw *Pterodactylus humeri* of up to 100 mm – but, sadly, not skull remains

We are glad the reviewer approves of our response. No further updates needed.

Much better now.

We are glad the reviewer approves of our change. No further updates needed.

Ok

We are glad the reviewer approves of our change. No further updates needed.

Perfect

We are glad the reviewer approves of our change. No further updates needed.

This was just a comment. Facts need not be changed.

We are glad the reviewer approves of our response. No further updates needed.

I know this paper. Still an open question that cannot be answered here.

This is more of a statement than a comment for clarification or improvement. As there is no published research on the relationship between enamel hardness and diet in bats, we feel it would be unwise to speculate on this in the manuscript.

I accept this.

We are glad the reviewer approves of our response. No further updates needed.

Reviewer 3

Remarks to the Author

I reviewed a previous iteration of this manuscript. It was excellent to begin with and the authors have satisfactorily addressed my concerns. I therefore have no issues with the present manuscript being accepted.

I now understand the justification for the species and specimens chosen for comparison. Thank you.

We are glad the reviewer approves of our change. No further updates needed.

Point taken about the benefits of PCA over DFA.

We are glad the reviewer approves of our response. No further updates needed.

I do think the results of the ancestral character estimations are clearer and more robust.

We are glad the reviewer approves of our changes. No further updates needed.

Thank you for the clarification regarding what is known and unknown regarding how non-occlusal wear is related to diet in non-mammals.

We are glad the reviewer approves of our response. No further updates needed.

Congratulations on the excellent paper. I look forward to seeing it in press.

Danielle Fraser, PhD

Canadian Museum of Nature